# NCP activates chloroplast transcription by controlling phytochrome-dependent dual nuclear and plastidial switches

Emily J. Yang [1,2,6,8], Chan Yul Yoo [1,8], Jiangxin Liu[3,4,8], He Wang[1], Jun Cao[5], Fay-Wei Li [2,7], Kathleen M. Pryer [2], Tai-ping Sun[2], Detlef Weigel[5], Pei Zhou [3] & Meng Chen [1]

Phytochromes initiate chloroplast biogenesis by activating genes encoding the photosynthetic apparatus, including photosynthesis-associated plastid-encoded genes (*PhAPGs*). *PhAPGs* are transcribed by a bacterial-type RNA polymerase (PEP), but how phytochromes in the nucleus activate chloroplast gene expression remains enigmatic. We report here a forward genetic screen in *Arabidopsis* that identified NUCLEAR CONTROL OF PEP ACTIVITY (NCP) as a necessary component of phytochrome signaling for *PhAPG* activation. NCP is dual-targeted to plastids and the nucleus. While nuclear NCP mediates the degradation of two repressors of chloroplast biogenesis, PIF1 and PIF3, NCP in plastids promotes the assembly of the PEP complex for *PhAPG* transcription. NCP and its paralog RCB are non-catalytic thioredoxin-like proteins that diverged in seed plants to adopt nonredundant functions in phytochrome signaling. These results support a model in which phytochromes control *PhAPG* expression through light-dependent double nuclear and plastidial switches that are linked by evolutionarily conserved and dual-localized regulatory proteins.

[1] Department of Botany and Plant Sciences, Institute for Integrative Genome Biology, University of California, Riverside, CA 92521, USA. [2] Department of Biology, Duke University, Durham, NC 27708, USA. [3] Department of Biochemistry, Duke University Medical Center, Durham, NC 27710, USA. [4] State Key Laboratory of Phytochemistry and Plant Resources in West China, Kunming Institute of Botany, Chinese Academy of Sciences, 650201 Kunming, China. [5] Department of Molecular Biology, Max Planck Institute for Developmental Biology, 72076 Tübingen, Germany. [6]Present address: Department of Pathology and Cell Biology, Columbia University Medical Center, New York, NY 10032, USA. [7]Present address: Boyce Thompson Institute, Ithaca, NY 14853, USA. [8]These authors contributed equally: Emily J. Yang, Chan Yul Yoo, Jiangxin Liu. Correspondence and requests for materials should be addressed to P.Z. (email: peizhou@biochem.duke.edu) or to M.C. (email: meng.chen@ucr.edu)

The control of organellar gene expression in the mitochondria and plastids is critical for cellular reprogramming in the eukaryotic cell. The regulation of gene activity in plastids is particularly important for plants because, although the vast majority of the genetic material of the ancestral cyanobacterial endosymbiont has been transferred to the nucleus[1], the plastid genome retains 100–120 genes encoding essential components of not only the plastidial transcriptional and translational machineries but also the photosynthetic apparatus[2]. The regulation of plastid-encoded photosynthesis-associated genes is pivotal for plants to establish photosynthetically active chloroplasts and thus is essential for plant survival.

Light is one of the most important environmental cues required for initiating chloroplast biogenesis in seed plants, including angiosperms (flowering plants) and some gymnosperms[3,4]. In dicotyledonous flowering plants, such as *Arabidopsis thaliana*, seedlings that germinate under the ground adopt a dark-grown developmental program called skotomorphogenesis, which promotes the elongation of the embryonic stem (hypocotyl) and inhibits leaf development and chloroplast biogenesis, a strategy that allows seedlings to emerge rapidly and easily from the soil. In darkness, the plastids in the leaf tissues differentiate into non-green, photosynthetically inactive etioplasts. Emerging into the sunlight triggers seedlings to transition to photomorphogenesis, which attenuates hypocotyl elongation and stimulates leaf development. The photomorphogenetic developmental program also enables chloroplast biogenesis and photosynthesis[5].

The transition to photomorphogenesis entails the massive transcriptional reprogramming of the nuclear genome initiated by photoreceptors, such as the red (R) and far-red (FR) photoreceptors, the phytochromes (PHYs), which play an essential role in chloroplast biogenesis[6–8]. The biological activity of PHYs can be turned on and off through light-dependent conformational switches between a R light-absorbing inactive Pr form and a FR light-absorbing active Pfr form[9]. In *Arabidopsis*, PHYs are encoded by five genes, *PHYA-E*, of which PHYA and PHYB are the predominant sensors of continuous FR and R light, respectively[5,10]. The earliest light response at the cellular level is the translocation of photoactivated PHYs from the cytoplasm to discrete subnuclear domains named photobodies[11]. PHYs bind directly to Phytochrome-Interacting Factors (PIFs) and colocalize with them on photobodies[12,13]. The PIFs are basic/helix-loop-helix transcription factors antagonistic to photomorphogenesis[14]. Most PIFs accumulate to high levels in dark-grown seedlings to promote hypocotyl elongation and inhibit chloroplast biogenesis by activating growth-relevant genes and repressing nuclear-encoded photosynthesis-associated genes, respectively[15,16]. PHYs repress the functions of PIFs by inhibiting their transcriptional activity and promoting their ubiquitin-and-proteasome-mediated degradation[13,14,17,18]. PHY-mediated PIF degradation is a central mechanism for inducing chloroplast biogenesis through the activation of photosynthesis-associated nuclear-encoded genes (*PhANGs*)[14]. The localization of PHYs to photobodies is closely associated with PIF3 degradation[12,13,17,19,20].

Light also induces the transcription of photosynthesis-associated plastid-encoded genes (*PhAPGs*)[21,22], which encode essential components of the photosynthetic apparatus, including the large subunit of the carbon fixation enzyme ribulose-1,5-bisphosphate carboxylase/oxygenase (rbcL) and the photosystem II reaction center D1 protein (psbA)[2]. Plastidial genes are transcribed by two types of RNA polymerases: a phage-type nuclear-encoded RNA polymerase (NEP) and a bacterial-type plastid-encoded RNA polymerase (PEP)[23]. While the NEP preferentially transcribes housekeeping genes, including plastid ribosomal RNAs and the core subunits of the PEP, the PEP mainly transcribes *PhAPGs*[24,25]. How PHYs in the nucleus control PEP-mediated *PhAPG* expression in plastids is largely unknown. A well-recognized challenge has been the lack of an efficient forward-genetic screening strategy that can distinguish chloroplast-deficient regulator mutants from other albino mutants with defects in genes encoding essential components of the chloroplast[26]. Our recent genetic studies of early PHY signaling have serendipitously uncovered a new class of photomorphogenetic mutants in *Arabidopsis* with a combination of albino and long-hypocotyl seedling phenotypes[19,27]. The founding member of this new mutant class, *hemera* (*hmr*), is defective in PHYB signaling and chloroplast biogenesis[19,28]. Albino mutants had been ignored previously in the context of light signaling because historically, chlorophyll-deficient mutants had been shown to retain normal PHY-mediated hypocotyl responses[29,30]. As a result, the entire class of tall-and-albino mutants like *hmr* had been overlooked[27]. We hypothesized that some of the tall-and-albino mutants might define missing components of PHY signaling for activating *PhAPG* expression. To test this hypothesis, we performed a forward genetic screen for tall-and-albino mutants. This screen identified Nuclear Control of PEP Activity (NCP), a dual-targeted nuclear/plastidial protein required for both the nuclear and plastidial signaling steps of *PhAPG* activation. We present evidence that NCP and its *Arabidopsis* paralog diverged in seed plants to adopt distinct roles in PHY signaling for *PhAPG* activation. We propose that PHYs control plastidial *PhAPG* expression via nucleus-to-plastid signaling, which comprises light-dependent double nuclear and plastidial switches that are governed by evolutionarily conserved and dual-localized regulatory proteins.

## Results

**Identification of *NCP*.** We performed a forward genetic screen in continuous monochromatic R light for mutants with a combination of tall and albino seedling phenotypes. The screen was conducted in the *PBG* (*PHYB-GFP*) background, a transgenic line in the null *phyB-5* background complemented with functional PHYB-GFP[31]. This screening strategy allowed us to assess whether the early signaling event of photobody formation is impaired in the mutants[19]. From 2,000 N-ethyl-N-nitrosourea or ethyl methanesulfonate mutagenized $F_2$ *PBG* families, we identified 23 tall-and-albino mutants. In this study, we focused on one of the mutants, which we named *ncp-1* (*nuclear control of PEP activity-1*) (Fig. 1a, b).

We used SHOREmap and mapped the mutation co-segregating with the tall-and-albino phenotype in *ncp-1/PBG* to a single G-to-A mutation in chromosome 2 at nucleotide 13,538,458, which results in a premature stop codon in gene *At2g31840* (Fig. 1c). Expressing the predicted cDNA of *At2g31840* under the constitutive *ubiquitin-10* promoter complemented the tall-and-albino phenotype of *ncp-1/PBG* (Supplementary Fig. 1a, b). We identified a second *ncp* allele in the Col-0 background, *ncp-10*, which carries a T-DNA insertion after nucleotide 13,538,811 in the second exon of *At2g31840* (Fig. 1c). The mRNA levels of *NCP* in *ncp-1/PBG* and *ncp-10* were more than 11-fold lower than those in their respective parental lines (Supplementary Fig. 1c). Both mutants are likely null alleles. Similar to *ncp-1/PBG*, *ncp-10* was tall and albino (Fig. 1d, e). Together, these results demonstrate that *At2g31840* is *NCP*.

*NCP* encodes a 350-amino-acid protein with a few recognizable motifs (Fig. 1c). Analysis by Phyre2 software (www.sbg.bio.ic.ac.uk/phyre2/) revealed a thioredoxin (Trx)-like domain (amino acid 212–319) at its C-terminus[32]. Interestingly, two subcellular targeting signals were found in NCP: an N-terminal transit peptide (amino acids 1–48) predicted by ChloroP[33] for chloroplast import and a nuclear localization signal (NLS) detected by NLS mapper[34] between amino acids 118 and 145

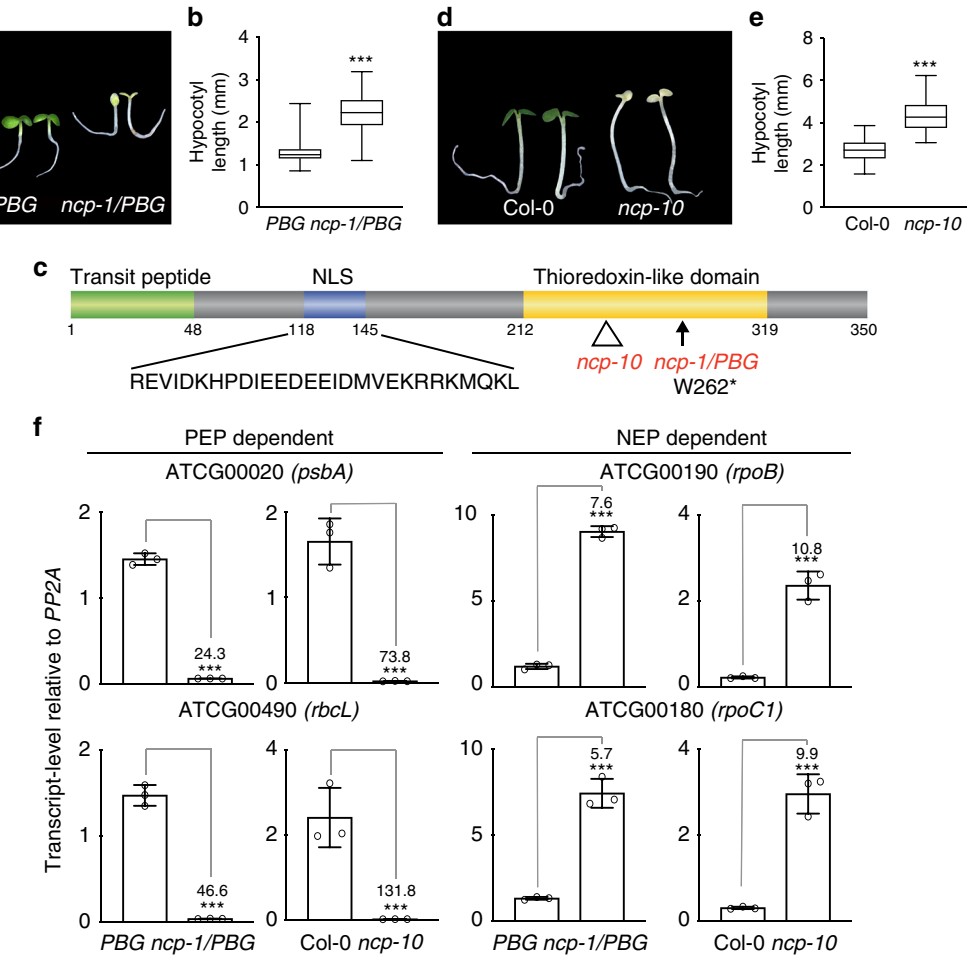

**Fig. 1** Identification of *NCP* by a screen for tall-and-albino mutants. **a** Representative images of 4-day-old *PBG* and *ncp-1/PBG* seedlings grown in 10 µmol m$^{-2}$ s$^{-1}$ continuous R light. **b** Box-and-whisker plots showing hypocotyl measurements of the seedlings in **a**. **c** Schematic illustration of the predicted domain structure of NCP. The mutation in *ncp-1/PBG* and the T-DNA insertion site in *ncp-10* are indicated. NLS, nuclear localization signal. **d** Representative images of 4-day-old Col-0 and *ncp-10* seedlings grown in 10 µmol m$^{-2}$ s$^{-1}$ continuous R light. **e** Box-and-whisker plots showing hypocotyl measurements of the seedlings in **d**. **f** qRT-PCR results showing the steady-state mRNA levels of the PEP-dependent *psbA* and *rbcL* and the NEP-dependent *rpoB* and *rpoC1* in 4-day-old *PBG*, *ncp-1/PBG*, Col-0, and *ncp-10* seedlings grown in 10 µmol m$^{-2}$ s$^{-1}$ continuous R light. Error bars represent SD of three biological replicates. The transcript levels were calculated relative to those of *PP2A*. The numbers above the right columns are the fold changes in gene expression between the columns. For the box-and-whisker plots in **b** and **e**, the boxes represent from the 25th to the 75th percentiles, and the bars equal the median values. For **b**, **e**, and **f**, asterisks (***) indicate a statistically significant difference between the values of the mutants and those of the wild-type or the parental line (Student's *t*-test, *p* ≤ 0.001). The source data of the hypocotyl measurements in **b**, **e** and the qRT-PCR data in **f** are provided in the Source Data file

(Fig. 1c). NCP has been identified previously as MRL7-L (Mesophyll-cell RNAi Library line 7-like)[35] and SVR4-like (Suppressor of Variegation4-like)[36] because of its essential role in chloroplast biogenesis, particularly for *PhAPG* activation[35,36]. However, the precise function of NCP in *PhAPG* regulation is still unknown. In agreement with published results, the expression of two PEP-dependent *PhAPG*s, *psbA* and *rbcL*, was downregulated by more than 24-fold and 73-fold in *ncp-1/PBG* and *ncp-10*, respectively, whereas the expression of NEP-dependent genes, such as *rpoB* and *rpoC1*, was upregulated by 5.7-fold to 10.8-fold (Fig. 1f). The divergent effects on PEP- and NEP-regulated genes are characteristics of mutants impaired specifically in the PEP function[24,28].

**NCP mediates phytochrome signaling.** To investigate the role of NCP in PHY signaling, we analyzed the hypocotyl elongation responses of the *ncp* mutants in continuous FR and R light to assess their effectiveness in PHYA and PHYB signaling, respectively[37]. These experiments showed that *ncp-10* and *ncp-1/PBG* were hyposensitive to R and FR light (Fig. 2a–d). The long hypocotyl phenotype of *ncp* relies on PHY signaling, as *ncp-10/phyB-9* and *ncp-10/phyA-211* double mutants were not taller than *phyB-9* and *phyA-211*, respectively (Fig. 2e–h). To further demonstrate NCP's role in PHY signaling, we crossed *ncp-1* to a constitutively active *phyB* allele *YHB*, which carries a Y276H mutation in PHYB's photosensory chromophore attachment domain that locks PHYB in an active form[38]. In the *ncp-1/YHB* double mutant, the constitutive photomorphogenetic phenotypes of *YHB* in the dark were partially suppressed (Fig. 2i, j), confirming that NCP is required for PHYB signaling. In contrast to the defects in response to FR and R light, *ncp-10* and *ncp-1/PBG* had normal hypocotyl responses in white and blue light (Supplementary Fig. 2), suggesting that NCP is not required for blue light signaling by cryptochromes. Together, these genetic data indicate that NCP participates specifically in PHY signaling.

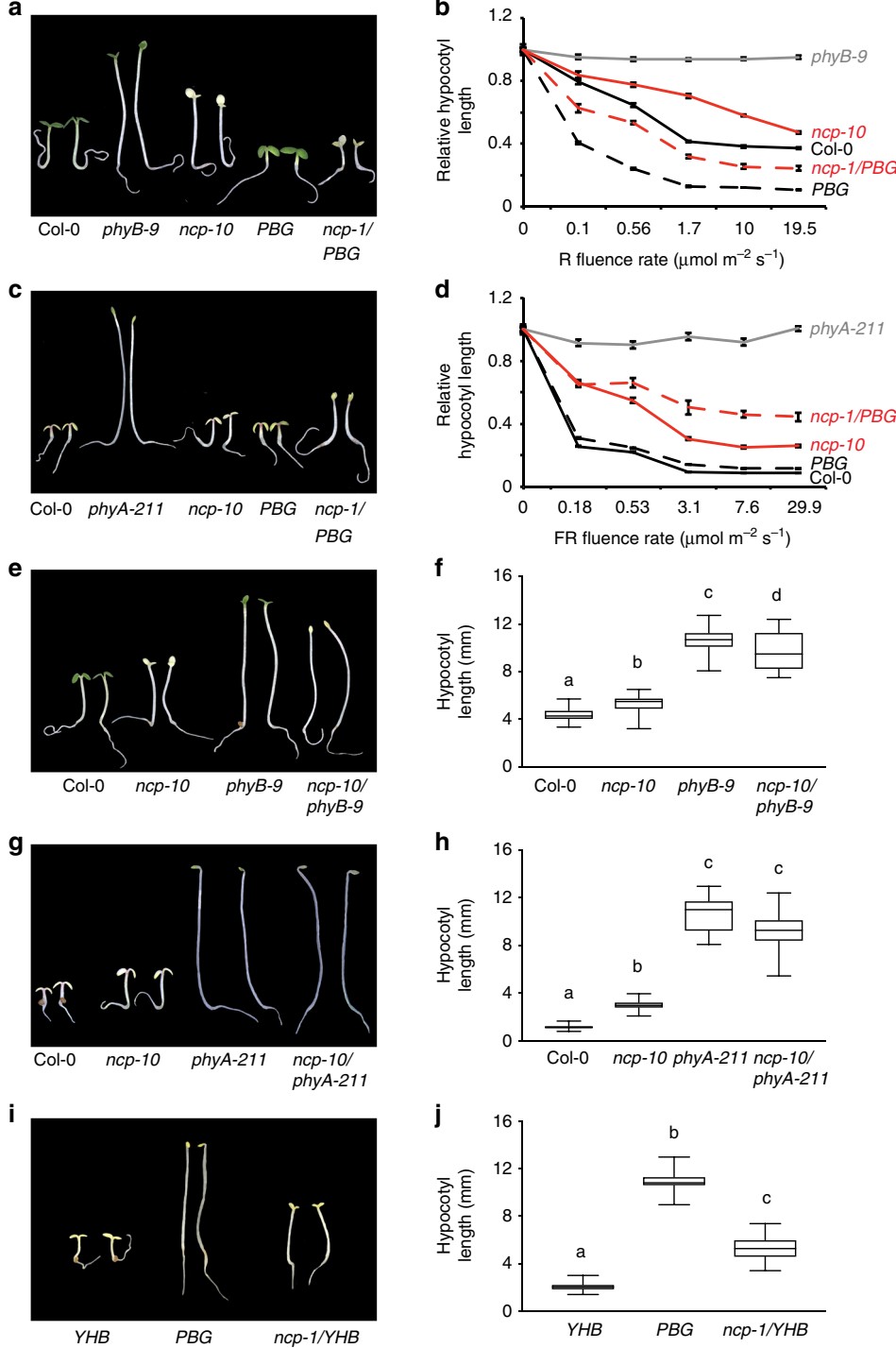

**Fig. 2** NCP mediates phytochrome signaling. **a** Representative images of 4-day-old Col-0, *phyB-9*, *ncp-10*, *PBG*, and *ncp-1/PBG* seedlings grown in 10 μmol m$^{-2}$ s$^{-1}$ R light. **b** R light fluence response curves showing the hypocotyl elongation responses of 4-day-old Col-0 (solid black line), *phyB-9* (solid gray line), *ncp-10* (solid red line), *PBG* (dotted black line), and *ncp-1/PBG* (dotted red line) seedlings grown in the dark and a series of R light intensities. **c** Representative images of 4-day-old Col-0, *phyA-211*, *ncp-10*, *PBG*, and *ncp-1/PBG* seedlings grown in 10 μmol m$^{-2}$ s$^{-1}$ FR light. **d** Fluence response curves for FR light showing the relative hypocotyl lengths of 4-day-old Col-0 (solid black line), *phyA-211* (solid gray line), *ncp-10* (solid red line), *PBG* (dotted black line) and *ncp-1/PBG* (dotted red line) seedlings grown in the dark and a series of FR light intensities. For **b** and **d**, error bars represent SE, and hypocotyl length in the light was calculated relative to that in the dark. **e**, Representative images of 4-day-old Col-0, *ncp-10*, *phyB-9*, and *ncp-10/phyB-9* seedlings grown in 10 μmol m$^{-2}$ s$^{-1}$ R light. **f** Box-and-whisker plots showing hypocotyl length measurements of the seedlings in **e**. **g** Representative images of 4-day-old Col-0, *ncp-10*, *phyA-211*, and *ncp-10/phyA-211* seedlings grown in 10 μmol m$^{-2}$ s$^{-1}$ FR light. **h** Box-and-whisker plots showing hypocotyl length measurements of the seedlings in **g**. **i** Representative images of 4-day-old dark-grown *YHB*, *PBG*, and *ncp-1/YHB* seedlings. **j** Box-and-whisker plots showing hypocotyl length measurements of the seedlings in **i**. For the box-and-whisker plots in **f**, **h**, and **j**, the boxes represent from the 25th to the 75th percentiles, and the bars equal the median values; samples with different letters show statistically significant differences in hypocotyl length (ANOVA, Tukey's HSD, $p \leq 0.001$, $n > 28$). The source data of the hypocotyl measurements in **b**, **d**, **f**, **h**, and **j** are provided in the Source Data file

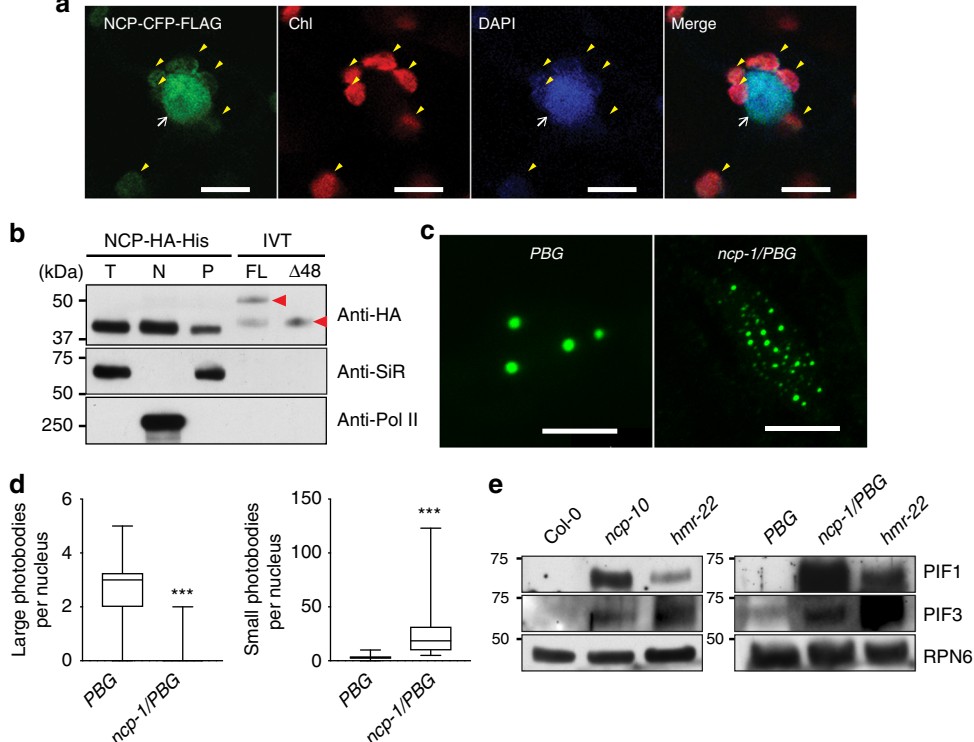

**Fig. 3** NCP participates in photobody biogenesis and PIF degradation in the nucleus. **a** Confocal images showing the subcellular localization pattern of transiently expressed NCP-CFP-FLAG in tobacco leaf cells. NCP-CFP-FLAG signals (green) were detected in chloroplasts (marked by yellow arrowheads) and the nucleus (indicated by a white arrow). The nucleus was labeled with DAPI. Chloroplasts were labeled with DAPI in the blue channel, as well as by chlorophyll autofluorescence in the red channel. Scale bars represent 10 μm. **b** Immunoblots showing NCP-HA-His from total, (T), nuclear (N), and plastidial (P) protein fractions of 2-day-old *NCP-HA-His/ncp-10* seedlings grown in 10 μmol m$^{-2}$ s$^{-1}$ R light. In vitro translated (IVT) NCP-HA-His and NCPΔ48-HA-His (indicated by red arrowheads) were used as molecular size controls. The HA-tagged NCP proteins were detected via anti-HA antibodies. Ferredoxin:sulfite reductase (SiR) and RNA Pol II were used as controls for the plastidial and nuclear fractions, respectively. **c** Confocal images of PHYB-GFP photobodies in epidermal cells from the top one third of the hypocotyls of 4-day-old *PBG* and *ncp-1/PBG* seedlings grown in 10 μmol m$^{-2}$ s$^{-1}$ R light. Scale bars represent 5 μm. **d** Box-and-whisker plots showing the numbers of large (>0.73 μm$^3$, left panel) and small (<0.73 μm$^3$, right panel) photobodies per nucleus in the epidermal cells of the top one third of the hypocotyls of *PBG* and *ncp-1/PBG* seedlings. The boxes represent from the 25th to the 75th percentiles, the whiskers extend to the maximum and minimum data points, and the center lines in each box indicate the median value. Asterisks (***) indicate a statistically significant difference between the value in *ncp-1/PBG* and that in *PBG* (Student's *t*-test, *p* ≤ 0.001). **e** Immunoblots showing the PIF1 and PIF3 levels in 4-day-old Col-0, *ncp-10, hmr-22, PBG,* and *ncp-1/PBG* seedlings grown in 10 μmol m$^{-2}$ s$^{-1}$ R light. RPN6 was used as a loading control. The source data for immunoblots in **b**, **e** and photobody analysis in **d** are provided in Source Data file

**NCP is a dual-targeted phytochrome signaling component**. The result that NCP participates in PHYA signaling and PHYB signaling, which occur mainly in the nucleus[39,40], contradicts the published data suggesting that NCP localizes only to plastids[35]. We therefore asked whether NCP, with a predicted NLS (Fig. 1c), is also targeted to the nucleus. We found that transiently expressed NCP tagged with CFP and FLAG (RCB-CFP-FLAG) in tobacco cells was dispersed in both chloroplasts and the nucleus (Fig. 3a). In agreement with this result, a functional HA-tagged and His-tagged NCP expressed in *ncp-10* (*NCP-HA-His/ncp-10*) was detected in the nuclear and plastidial protein fractions (Fig. 3b and Supplementary Fig. 3). Surprisingly, although plastidial NCP is expected to be smaller than nuclear NCP due to the removal of its transit peptide during plastid import, the total, nuclear, and plastidial fractions of NCP-HA-His had similar molecular masses (Fig. 3b). To further examine the size of NCP-HA-His, we ran side-by-side in vitro translated full-length NCP-HA-His and the predicted plastidial NCP without the N-terminal 48 amino acids (NCPΔ48-HA-His). NCP-HA-His in vivo was significantly smaller than the in vitro translated full-length NCP-HA-His and similar to NCPΔ48-HA-His (Fig. 3b). These results indicate that NCP is dual-targeted to plastids and the nucleus and

imply that NCP might localize to the plastids first and then translocate to the nucleus similar to HMR[41].

To understand how NCP participates in PHY signaling, we asked whether NCP is required for the earliest light response, PHYB localization to photobodies[11]. In *PBG* seedlings grown under 10 μmol m$^{-2}$ s$^{-1}$ R light, PHYB-GFP localized to a few large photobodies (Fig. 3c, d)[19,42]. In striking contrast, PHYB-GFP localized mostly to small photobodies in *ncp-1/PBG* (Fig. 3c, d). We then tested if NCP is required for the PHY-mediated degradation of the antagonistic transcription factors of PHY signaling, the PIFs, because PIF degradation is closely associated with PHYB localization to large photosbodies[19,42]. Intriguingly, the two well-characterized light-labile PIFs, PIF1, and PIF3[14], accumulated or failed to be completely degraded in light-grown *ncp-1/PBG* and *ncp-10* (Fig. 3e). Together, these results demonstrate that NCP participates in the early light signaling events of photobody biogenesis and the degradation of PIF1 and PIF3.

**NCP activates *PhAPGs* in the nucleus and plastids**. The PEP forms multisubunit protein complexes consisting of the bacterial-

type core subunits and plant-specific PEP-associated proteins[28,43]. We have shown recently that *PhAPG*s are activated by a linked nuclear and plastidial dual-switching mechanism in which PHY-mediated PIF degradation in the nucleus triggers the assembly of the PEP into a 1000-kDa complex in plastids for *PhAPG* transcription[44]. With four *PIF* genes knocked out, *PIF1, PIF3, PIF4*, and *PIF5*, the *pifq* mutant could trigger PEP assembly and *PhAPG* activation in the dark[44]. The dual nuclear-and-plastidial localization of NCP raised the question of whether NCP only functions in promoting PIF degradation in the nucleus or also regulates PEP assembly and activation directly in plastids. To distinguish between these two possibilities, we generated a *ncp-10/pif1/pif3/pif4/pif5 (ncp-10/pifq)* quintuple mutant. We reasoned that if NCP activates *PhAPG* expression mainly by promoting PIF degradation in the nucleus, removing the four *PIF*s in *ncp-10* should rescue its albino phenotype. The *ncp-10/pifq* mutant largely rescued the long-hypocotyl phenotype of *ncp-10* (Fig. 4a). However, the *ncp-10/pifq* mutant was slightly but significantly taller than *pifq* (Fig. 4a, b), which could be due to NCP-dependent regulation of other PIFs or a PIF-independent retrograde signaling from the defective chloroplasts, as *ncp-10/pifq* remained albino[45]. The expression of PEP-dependent *PhAPG*s was still impaired in *ncp-10/pifq*, while the expression of NEP-dependent plastidial genes was elevated (Fig. 4c). We resolved the PEP complex from *Arabidopsis* using blue-native gel electrophoresis and monitored its size by immunoblotting using antibodies against either the core β subunit, rpoB, or one of the PEP-associated proteins, HMR/pTAC12[19,28,43]. We found that the PEP failed to form a 1000-kDa complex in *ncp-10* (Fig. 4d), indicating that NCP is required for PEP assembly. The defect in PEP assembly was not rescued in *ncp-10/pifq* (Fig. 4d). Together, these results indicate that in addition to its nuclear function in PIF degradation, NCP also facilitates PEP assembly directly in plastids.

**NCP and its paralog *RCB* diverged in seed plants.** *NCP* has a paralog in *Arabidopsis*, *At4g28590*, which we named *Regulator of Chloroplast Biogenesis (RCB)*[44]. Previous studies have shown that *NCP* and *RCB* are present in angiosperms[35]. However, only one copy of *NCP*-like gene was found in the genomes of non-flowering plants, such as the moss *Physcomitrella patens* and lycophyte *Selaginella moellendorffi*[35,46]. Because these analyses did not include other seed plants, like gymnosperms, or the lineages between seed plants and lycophytes, like ferns, it remains unclear when *NCP* and *RCB* diverged during the evolution of land plants. We therefore searched for *NCP*-like sequences in all major land plant lineages, including bryophytes, lycophytes, ferns, and seed plants, and utilized Randomized Axelerated Maximum Likelihood (RAxML) to construct a phylogenetic tree with bootstrapping[47]. The resulting phylogeny revealed a single copy of *NCP-like* genes in ferns, lycophytes, and bryophytes, whereas the seed plants, including both angiosperms and gymnosperms, contain *NCP* and *RCB* (Fig. 5a). *NCP*-like sequences have not been identified in prokaryotic photosynthetic organisms, such as *Rhodobacter sphaeroides* and *Synechocystis sp.* PCC 6803, or in the single-cell green alga *Chlamydomonas reinhardtii*[35,46]. Consistent with these results, we did not find *NCP* homologs in the algal genomes of *Klebsormidium flaccidum*[48] and *Micromonas pusilla CCMP1545*[49]. These results indicate that an *NCP*-like gene emerged in early land plants and duplicated and diverged into *NCP* and *RCB* in seed plants.

Intriguingly, we also identified RCB from the screen for tall-and-albino mutants[44]. We showed that RCB is dual-localized to the plastids and the nucleus[44]. However, different from NCP, plastidial RCB does not play an essential role in chloroplast

biogenesis[44]. Instead, RCB initiates chloroplast biogenesis primarily in the nucleus to promote PIF1 and PIF3 degradation (Fig. 5d)[44]. These results indicate that NCP and RCB have adopted non-redundant roles in regulating chloroplast biogenesis. The *ncp-10/rcb-10* double mutant had long hypocotyl and albino phenotypes that were similar to those of the single mutants (Fig. 5b, c), suggesting that NCP and RCB function in the same PHY-dependent pathway for initiating chloroplast biogenesis. We propose that *NCP* and *RCB* diverged to evolve distinct regulatory roles in PHY signaling likely to accommodate the regulation of chloroplast biogenesis by light in seed plants.

**NCP and RCB contain a non-catalytic thioredoxin-like domain.** NCP and RCB possess a C-terminal Trx-like domain. Trx is a small redox-active protein with a universally conserved dithiol (-Cys-X-X-Cys-) active site in which the Cys residues provide the sulfhydryl groups required for reducing disulfide bonds in target proteins[50]. Interestingly, neither NCP nor RCB contains the -Cys-X-X-Cys- catalytic motif (Fig. 6a). Surprisingly, a previous study suggested that RCB had Trx activity in vitro[51]. To seek a structural explanation for the Trx reductase activity, we solved the NMR structure of NCP's Trx-like domain. The Trx-like domain of NCP displays a typical Trx-like fold—a five-stranded β-sheet with a β1-β5 arrangement surrounded by four α-helixes (Fig. 6b and Supplementary Figs. 4–6). A stereo view of the NMR structural ensemble of the NCP Trx-like domain is shown in Supplementary Fig. 7, and the detailed statistics on the structural ensemble are given in Table 1. Based on the structure of NCP, we modeled the structure of the Trx-like fold of RCB (Fig. 6c). The structure of NCP's Trx-like domain overlays nicely with that of *E. coli* Trx (Fig. 6d)[52]. The Trx-like domains of NCP and RCB exhibit the same βαβαβαββα secondary structural arrangement as those of *E. coli* Trx (Fig. 6a–d) but without a -Cys-X-X-Cys- catalytic motif. Therefore, the structural data do not support a Trx reductase activity. We then performed the insulin reduction assay using *E. coli* expressed recombinant NCP and RCB fragments (Fig. 6e)[53]. These experiments did not detect any Trx reductase activity in NCP or RCB (Fig. 6f, g). Based on the structural and biochemical analyses, we conclude that RCB and NCP contain a non-catalytic Trx-like domain.

## Discussion

PHY signaling initiates chloroplast biogenesis by activating photosynthesis-associated genes encoded by the nuclear and plastidial genomes. PHYs regulate nuclear gene expression by directly modulating the activity and stability of transcription factors in the nucleus[14,27]. However, how PHYs in the nucleus control gene expression in plastids, particularly the activation of *PhAPG*s by the PEP, remains poorly understood. Here we report the identification of a dual-targeted nuclear/plastidial signaling component, NCP, which participates in both nuclear and plastidial PHY signaling events for *PhAPG* activation (Fig. 5d). Our results provide evidence supporting the model that PHYs control plastidial gene expression through dual nuclear and plastidial switches, which are governed by evolutionarily conserved dual-targeted regulatory proteins (Fig. 5d).

It has been proposed for decades that plastid-encoded genes are controlled by the cell nucleus through anterograde nucleus-to-plastid signaling[26]. A well-recognized challenge, which hindered the identification of such an anterograde signaling pathway, had been the lack of an effective forward-genetic screening strategy for distinguishing chloroplast-deficient regulator mutants from other albino mutants with defects in genes encoding essential components of the chloroplast[26]. Different from all previous genetic screens, we searched for mutants with a

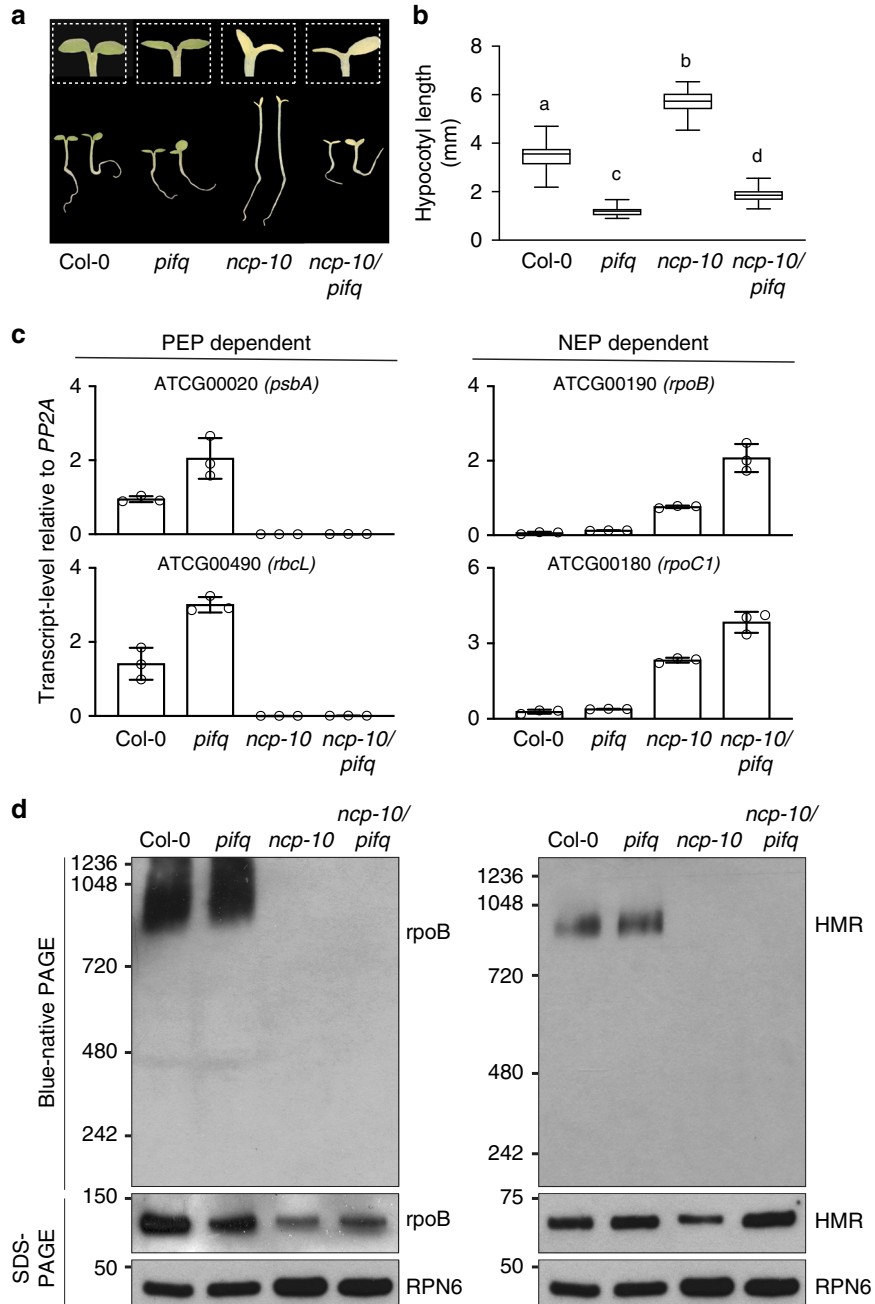

**Fig. 4** NCP promotes the assembly and activation of the PEP in plastids. **a** A *ncp-10/pifq* mutant rescues the long hypocotyl phenotype, but not the albino phenotype, of *ncp-10*. Left panel: representative images of 4-day-old Col-0, *pifq*, *ncp-10*, and *ncp-10/pifq* seedlings grown in 10 μmol m⁻² s⁻¹ R light. Inlets show magnified images of embryonic leaves of one of the corresponding seedlings below. **b** Box-and-whisker plots showing hypocotyl length measurements for the seedlings in **a**. The boxes represent from the 25th to the 75th percentiles, and the bars represent the median values. Samples with different letters have statistically significant differences in hypocotyl length (ANOVA, Tukey's HSD, $p \leq 0.001$, $n > 30$). **c** qRT-PCR analyses of the transcript levels of selected PEP-dependent and NEP-dependent genes in 4-day-old Col-0, *pifq*, *ncp-10*, and *ncp-10/pifq* seedlings grown under 10 μmol m⁻² s⁻¹ R light. Transcript levels were calculated relative to those of *PP2A*. Error bars represent SD of three biological replicates. **d** Immunoblots showing the level of the PEP complex (blue-native PAGE), as well as the levels of rpoB and HMR proteins (SDS-PAGE) in 4-day-old Col-0, *pifq*, *ncp-10*, and *ncp-10/pifq* seedlings grown in 10 μmol m⁻² s⁻¹ R light using antibodies against rpoB or HMR. RPN6 was used as a loading control. The source data underlying the hypocotyl measurements in **b**, the qRT-PCR data in **c**, and the immunoblots in **d** are provided in the Source Data file

combination of long hypocotyl and albino phenotypes, which is indicative of defects in nuclear PHY signaling and chloroplast biogenesis[17,19]. This forward genetic screen identified NCP and its paralog, RCB[44]. Our investigation of RCB revealed that PHYs induce *PhAPG* transcription through a nucleus-to-plastid anterograde signaling pathway linking two required switching mechanisms: PHY-mediated PIF degradation in the nucleus and the assembly of the PEP into a 1000-kDa complex in plastids (Fig. 5d)[44]. Interestingly, although RCB is dual-targeted to plastids and the nucleus, it activates *PhAPG* expression primarily in the nucleus by promoting PIF1 and PIF3 degradation[44]. An *rcb-10/pifq* mutant, in which four nuclear PIF transcription factors

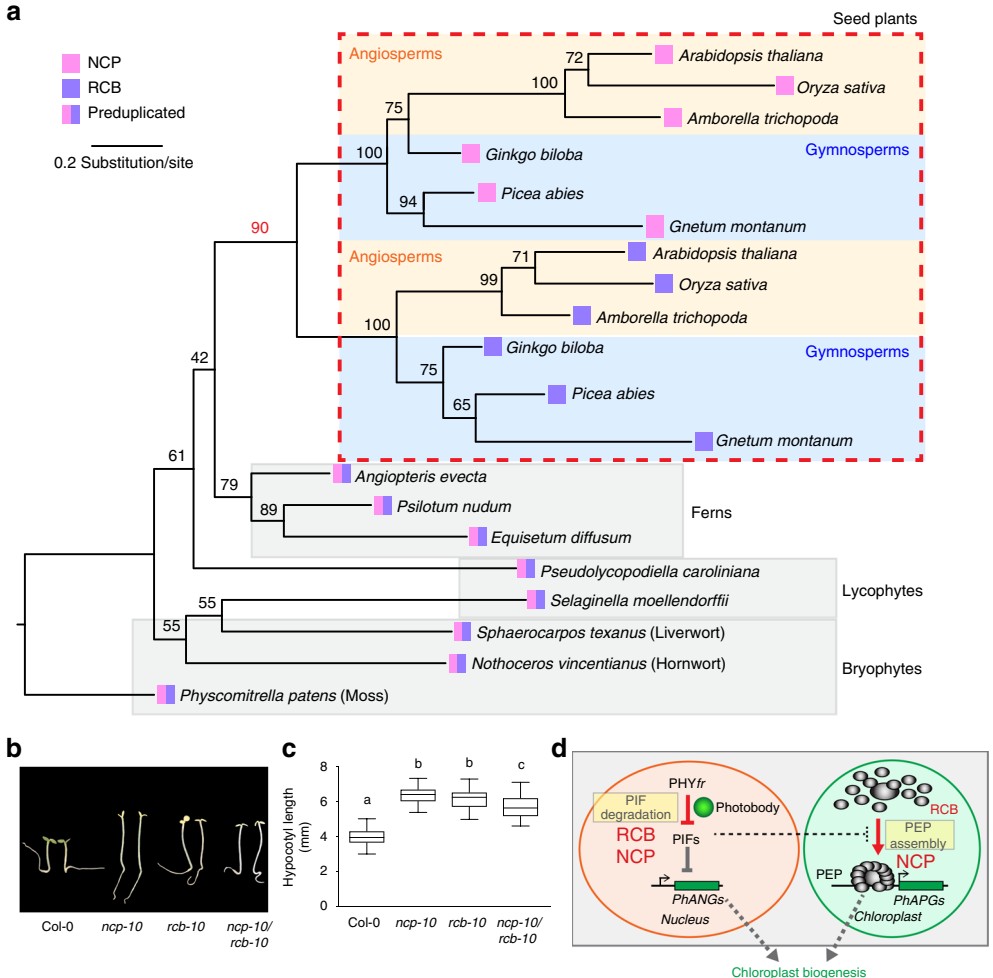

**Fig. 5** *NCP* and *RCB* diverged in seed plants to adopt distinct roles in PHY signaling. **a** Maximum likelihood phylogenetic tree of NCP and RCB from seed plants, ferns, lycophytes, and bryophytes. The gene IDs of the sequences used for the phylogenetic analysis are listed in Supplementary Table 4. The phylogenetic tree was constructed using the Randomized Axelerated Maximum Likelihood method. The numbers at each node indicate the bootstrap value (%) from 1000 replications. The length of the branches represents the extent of divergence based on the scale at the bottom. **b** Representative images of 4-day-old Col-0, *ncp-10*, *rcb-10*, and *ncp-10/rcb-10* seedlings grown in 10 µmol m$^{-2}$ s$^{-1}$ R light. **c** Box-and-whisker plots showing hypocotyl length measurements of the seedlings in **b**. The boxes represent from the 25th to the 75th percentiles, and the bars represent the median values. Samples with different letters exhibit statistically significant differences in hypocotyl length (ANOVA, Tukey's HSD, $p \leq 0.001$, $n > 40$). **d** Model for the PHY-mediated nucleus-to-plastid light signaling pathway for *PhAPG* activation. PHYs activate *PhAPGs* through a linked nuclear and plastidial dual-switching mechanism in which PHY-mediated PIF degradation in the nucleus triggers the assembly and activation of the PEP in plastids for *PhAPG* transcription. PIF degradation is the nodal signaling step governing the activation of *PhAPGs* and photosynthesis-associated nuclear-encoded genes (*PhANGs*). While RCB participates primarily in the nuclear switching mechanism, NCP controls both the nuclear and plastidial switches. The source data underlying the hypocotyl measurements in **c** are provided in the Source Data file

are removed, rescues *rcb-10*'s defects in PEP assembly, *PhAPG* activation, and chloroplast biogenesis[44]. In contrast to RCB, NCP is required for controlling both the nuclear and plastidial switches for *PhAPG* activation. Although NCP had been reported previously for its essential role in chloroplast biogenesis[35,36], the function of NCP in chloroplast biogenesis was still unknown. Different from the current view that NCP localizes only to plastids[35], we show that NCP is dual-targeted to the plastids and the nucleus. The dual localization of NCP is supported by transiently expressed NCP-CFP-FLAG (Fig. 3a) and subcellular fractionation results using a *NCP-HA-His/ncp-10* transgenic line (Fig. 3b). Although it is possible that the nuclear localization of NCP in these experiments could be due to overexpression of NCP, this is highly unlikely because a direct role of NCP in nuclear PHY signaling is also supported by the overwhelming genetic evidence. NCP is required for both PHYA and PHYB

signaling (Fig. 2). RCB participates in the early light signaling events of photobody biogenesis (Fig. 3c, d). Moreover, both PIF1 and PIF3 accumulate in *ncp-10* in the light (Fig. 3e), and the long hypocotyl phenotype of *ncp-10* was rescued in *ncp-10/pifq* mutant (Fig. 4a), further supporting the notion that NCP is directly involved in PIF degradation in the nucleus. Intriguingly, the *ncp-10/pifq* mutant does not rescue the *ncp-10*'s defects in PEP assembly, *PhAPG* activation, and chloroplast biogenesis (Fig. 4). Together, these results indicate that NCP facilitates both PHY-mediated PIF degradation in the nucleus, possibly by promoting photobody biogenesis, as well as the assembly of the PEP in plastids (Fig. 5d).

We show that seed plants, including angiosperms and gymnosperms, contain both NCP and RCB (Fig. 5a). In non-seed land plants, including ferns and mosses, there is only one copy of *NCP-like* gene. It is intriguing that seed plants have evolved both

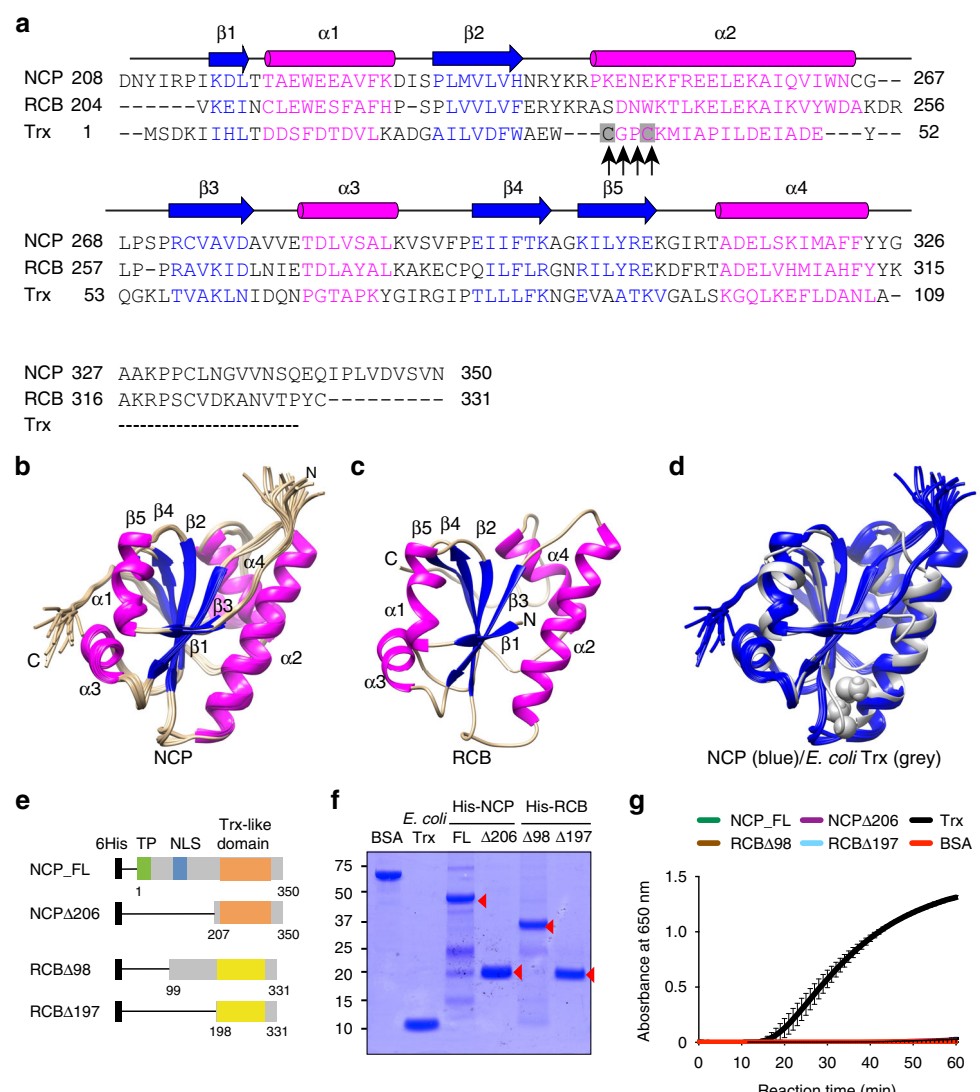

**Fig. 6** NCP and RCB structurally resemble *E. coli* Trx but lack reductase activity. **a** Sequence alignment of the Trx-like domains of NCP and RCB with that of *E. coli* Trx. Magenta characters represent alpha helices and blue characters represent beta sheets. The black arrows indicate the conserved catalytic -Cys-X-X-Cys- motif in *E. coli* Trx. **b** NMR structures of NCP's Trx-like domain (PDB ID: 6NE8). **c** Simulated structure of RCB's Trx-like domain based on the NMR structure of NCP. **d** Overlay of the structure of NCP's Trx-like domain with the crystal structure of *E. coli* Trx (PDB ID: 1XOB). The two cysteines in *E. coli* Trx are shown. **e** Schematic illustration of NCP and RCB fragments used for the in vitro Trx reductase assays. **f** SDS-PAGE gels showing BSA (negative control), *E. coli* Trx (positive control), and purified His-tagged recombinant NCP and RCB fragments. **g** Trx activity assays showing that recombinant NCP and RCB proteins lack Trx activity. Trx activity was determined based on the increase in turbidity after the reduction of insulin, as measured by the absorbance at 650 nm[53]. *E. coli* Trx and BSA proteins were used as a positive control and a negative control, respectively. The source data of the Trx activity assay in **g** are provided in the Source Data file

NCP and RCB, which play non-redundant roles in PHY signaling (Fig. 5b, c). One possibility is that having both NCP and RCB can provide an additional regulatory mechanism for the light-dependent chloroplast biogenesis or greening in seed plants[3,4]. NCP and RCB might work in concert in the nucleus to promote PHYs to localize to photobodies for PIF degradation; for example, they could form a heteromeric protein complex. We will test this hypothesis in future investigations. The functions of NCP and RCB in plastids are likely distinct. Only NCP plays an essential role in PEP assembly and *PhAPG* activation in plastids (Fig. 4)[44]. NCP and RCB show different localization patterns in plastids: while RCB is localized to the nucleoid, NCP is localized to the stroma (Fig. 3a)[35]. It is unclear how the localization patterns of these proteins relate to their platidial functions. Our previous work has identified another dual-targeted nuclear and plastidial protein, HMR, which also participates in the PHY-mediated

control of *PhAPG* activation[19,28,41]. While nuclear HMR acts as a transcriptional activator interacting directly with PIFs to mediate PIF1 and PIF3 degradation, plastidial HMR, also called pTAC12, is an essential component of the PEP complex[28,54,55]. Our genetic studies have so far identified three dual-targeted nuclear/plastidial molecules in PHY signaling—HMR[19,41], RCB[44], and NCP (this study). One pressing upcoming task is to determine the regulation and mechanism of their dual localization, as well as to understand the significance of their dual-localization in PHY signaling and in nucleus-plastid communication.

NCP and RCB each contain a Trx-like domain. Although a previous study suggested that RCB's Trx domain possesses Trx reductase activity[51], our results demonstrate convincingly that the Trx-like domains of NCP and RCB are structurally similar to that of *E. coli* Trx but lack reductase activity (Fig. 6a–c). Neither NCP nor RCB contains the universally conserved -Cys-X-X-Cys-

### Table 1 Structural statistics for the Trx-like domain of NCP

| NCP Trx-like domain (D208-N350)[a] | |
| --- | --- |
| NOE distance restraints | 6313 |
| Short-range ($\lvert i - j \rvert \leq 1$) | 2118 |
| Medium-range ($1 < \lvert i - j \rvert \leq 4$) | 1457 |
| Long-range ($\lvert i - j \rvert \geq 5$) | 2738 |
| Dihedral angle constraints[c] | 245 |
| Target function value | 2.34 ± 0.05 |
| Ramachandran plot[d] | |
| Favored region (98%) | 88% |
| Allowed region (>99.8%) | 12% |
| Mean pairwise RMSD (RCBL Y210-V347) | |
| Backbone | 0.29 ± 0.03 Å |
| Heavy atoms | 0.65 ± 0.03 Å |

[a]None of these structures exhibit distance violations greater than 0.4 Å or dihedral angle violations greater than 4°
[b]Two constraints ($d_{HN-O} \leq 2.5$ Å and $d_{N-O} \leq 3.5$ Å) are used for each identified hydrogen bond
[c]Dihedral angle constraints were generated by TALOS+ based on backbone atom chemical shifts[76], and by analysis of NOE patterns
[d]MOLPROBITY was used to assess the quality of the structures[78]

catalytic motif required for reductase activity (Fig. 6d)[50]. Trx could participate in a regulatory mechanism independent of redox activity, depending on the ability to interact with other proteins to form functional protein complexes. For example, *E. coli* Trx that lacks the catalytic cysteine residues in its active site can enhance the processivity of the bacteriophage T7 DNA polymerase[56,57]; the active site in this case mediates the interaction with the T7 DNA polymerase[58]. Similarly, RCB and NCP could use their non-catalytic Trx-like domains for protein-protein interactions. Future investigations will test this hypothesis to determine the biochemical functions of NCP and RCB in nuclear PHY signaling as well as PEP assembly and activation.

## Methods

**Plant materials and growth conditions**. The *PBG* line in the Landsberg *erecta* (L*er*) background has been previously reported[31]. The *ncp-1/PBG* mutant line was isolated from the tall-and-albino mutant screen and backcrossed to *PBG* three times. The *ncp-10* mutant (Col-0 background) was identified from GABI-Kat T-DNA insertion line GK-518H02[59] and obtained from the *Arabidopsis* Biological Resource Center (CS449718). *ncp-1* was genotyped using a dCAPS (derived Cleaved Amplified Polymorphic Sequences) marker using the following PCR primers: forward: agagaaggcgattcaagtcatat, reverse: ctggaagtaataatgacccag. NdeI digestion of the PCR product yields 129-bp and 20-bp fragments for L*er* and one 149-bp fragment for *ncp-1*. The *Arabidopsis* mutants *phyB-9*, *phyA-211*, and *pifq*, all in the Col-0 background, were used for the physiological studies. *YHB*, a constitutively active *phyB* mutant in the L*er* background has been previously reported[38]. Seeds were surface-sterilized and plated on half-strength Murashige and Skoog (MS) growth medium with Gamborg's vitamins (MSP0506, Caisson Laboratories, North Logan, UT) containing 0.5 mM MES pH 5.7 and 0.8% agar (w/v) (A038, Caisson Laboratories, North Logan, UT)[19]. Seeds were stratified in the dark at 4 °C for 5 days. Seedlings were grown at 21 °C in an LED chamber (Percival Scientific, Perry, IA) under the indicated light conditions. Fluence rates of light were measured using an Apogee PS200 spectroradiometer (Apogee instruments Inc., Logan, UT) and SpectraWiz software (StellarNet, Tampa, FL).

**Hypocotyl length measurement**. For the measurement of hypocotyl length, 4-d-old seedlings grown under different light conditions were scanned using an Epson Perfection V700 photo scanner, and hypocotyl lengths were measured using NIH ImageJ software (https://imagej.nih.gov/ij/). Box-and-whisker plots of hypocotyl measurements were generated using Prism 7 software (GraphPad, San Diego, CA). Images of representative seedlings were captured using a Leica MZ FLIII stereo microscope (Leica microsystems Inc., Buffalo Grove, IL) and processed using Adobe Photoshop CC (Adobe Systems, Mountain View, CA).

**Mutant generation**. *PBG* seeds were hydrated in 45 ml of ddH$_2$O with 0.005% Tween-20 for 4 h and washed with ddH$_2$O twice. The washed seeds were soaked in 1 mM N-ethyl-N-nitrosourea or ethyl methanesulfonate solution for 15 h with rotation. Then, the seeds were thoroughly washed in ddH$_2$O, plated on MS growth media, and stratified at 4 °C for 4 days. The M1 seedlings were transferred to soil, and the M2 seeds were collected from individual M1 plants. The M2 generation was screened under monochromatic R light.

**Genetic Mapping via SHOREmap**. *ncp-1/PBG* (L*er*) was crossed to Col-0 to generate an F2 mapping population. Genomic DNA from pools of more than 800 F2 seedlings with a tall-and-albino phenotype was extracted as follows[60]. Seedlings were ground in liquid nitrogen and resuspended in nuclear extraction buffer containing 10 mM Tris-HCl pH 9.5, 10 mM EDTA pH 8.0, 100 mM KCl, 500 mM sucrose, 4 mM spermidine, 1 mM spermine, and 0.1% β-mercaptoethanol. Two mililiter of lysis buffer containing 10% Triton X-100 in nuclei extraction buffer was added into the homogenized tissues. After incubation on ice for 2 min, the homogenate was centrifuged at 2000 × *g* at 4 °C for 10 min. The nuclei pellet was resuspended in 500 μl of CTAB buffer containing 100 mM Tris-HCl pH 7.5, 0.7 M NaCl, 10 mM EDTA pH 8.0, 1% CTAB, and 1% β-mercaptoethanol. After incubation at 60 °C for 30 min, genomic DNA was extracted with chloroform/isoamyl alcohol (24:1) and precipitated with isopropanol by centrifugation at 20,000 × *g* at 4 °C for 10 min. Illumina paired-end libraries with 300-bp insert sizes were constructed per the manufacturer's instructions. Eighty-base paired-end reads were generated on an Illumina Genome Analyzer II, targeting approximately 25× genome coverage. The polymorphisms, including SNPs, indels up to 3 bp, and large deletions, were identified using SHOREmap[61]. Genomic regions enriched for mutant parental markers were identified with SHOREmap[62,63]. Variants in the final mapping interval that were absent from the L*er* background and that were predicted to have a large impact on ORF integrity were prioritized as candidate mutations.

**Plasmid construction and generation of transgenic plants**. All the primers used for plasmid construction are listed in Supplementary Table 1. The *pCHF1-UBQ10p::NCP-HA-His* construct was generated by cloning the *UBQ10* promoter and the full-length coding sequence of *NCP* into the EcoRI and PstI sites of *pCHF1-HA-His* vector using Gibson assembly (New England Biolabs, Ipswich, MA); the construct was prepared by inserting a DNA fragment encoding (PT)4P-3HA-6His into the PstI and SalI sites of the *pCHF1* vector[64]. Transgenic lines were generated by transforming *ncp-10* heterozygous plants with *Agrobacterium tumefaciens* strain GV3101 containing the *pCHF1-UBQ10::NCP-HA-His* construct. The T1 transgenic plants were selected on half-strength MS medium containing 100 μg/ml gentamycin and screened for transgenic plants with a homozygous *ncp-10* mutation. The T2 lines with a single-locus insertion status of the transgene were selected based on a 3:1 segregation ratio for gentamycin resistance. The T3 generation plants homozygous for the transgene were used for the experiments.

The *NCP-CFP* construct used for the tobacco transient expression assay was generated by amplifying the coding sequences of *NCP* and *CFP-FLAG* and inserting them into the XmaI and XbaI sites of the *pCHF3* vector. The *NCP-CFP* construct was transformed into *Agrobacterium tumefaciens* strain GV3101 for the transient expression assay. The constructs used for in vitro transcribed and translated HA-and His-tagged NCP or NCPΔ48 (deletion of N-terminal 48 amino-acid transit peptide) was generated by amplifying the *NCP-HA-His* fragments and inserting them into EcoRI and BamHI sites of the *pCMX-PL2* vector.

The constructs used for expressing N-terminally His-tagged NCP and RCB proteins in *E. coli* were made in the *pET15b* and *pET28a* vectors, respectively. NCP or NCPΔ206 (Trx-like domain only, aa 207–350) were amplified by PCR and ligated into the NdeI and XhoI sites of the *pET15b* vector using T4 DNA ligase. RCBΔ98 (aa 99–331) and RCBΔ197 (Trx-like domain only, aa 198–331) were amplified and inserted into the BamHI and HindIII sites of the *pET28a* vector using Gibson assembly.

**RNA extraction and quantitative PCR**. Total RNA from seedlings of the indicated genotypes and growth conditions was isolated using a Quick-RNA MiniPrep Kit with on-column DNase I treatment (Zymo Research, Irvine, CA). cDNA was synthesized using Superscript II First-strand cDNA Synthesis Kit (ThermoFisher Scientific, Waltham, MA). Oligo(dT) primers were used for the analysis of nuclear gene expression, and a mixture of oligo(dT) and plastidial-gene-specific primers was used for the analysis of plastidial genes. qRT-PCR was performed with FastStart Universal SYBR Green Master Mix on a LightCycler 96 System (Roche, Basel, Switzerland). The transcript level of each gene was normalized to that of *PP2A*. All primers used for qRT-PCR and cDNA synthesis are listed in Supplementary Tables 2 and 3.

**Phylogenetic analysis**. We acquired the *NCP*-like sequences of seed plants (*Arabidopsis thaliana, Oryza sativa, Amborella trichopoda, Ginkgo biloba, Picea abies, Gnetum montanum*) with implemented BLASTp from Phytozome or individual genome portals[65,66]. We obtained the *NCP*-like sequences in ferns (*Angiopteris evecta, Psilotum nudum, Equisetum diffusum*), lycophytes (*Pseudolycopodiella caroliniana, S. moellendorffii*), and bryophytes [*Nothoceros vincentianus* (a hornwort), *Sphaerocarpos Texanus* (a liverwort), and *Physcomitrella patens*] from the One Thousand Plants Project (www.onekp.com)[67] using transcriptome mining based on the BlueDevil Python pipeline[68]. The homologs of NCP and RCB from 6 representative seed plants were BLASTed and obtained from Phytozome[65], Congenie (congenie.org)[66], and the Amborella Genome Database (www.amborella.org)[69] (Supplementary Table 4). The homologous sequences from ferns, bryophytes, and lycophytes were mined using the Python pipeline BlueDevil[68] from transcriptomes generated from One

Thousand Plants Project[67]. The nucleotide sequences obtained from the transcriptomes were translated into amino acid sequences, including 20 sequences from 14 species. The sequences were aligned using MUSCLE[70]. To decrease ambiguities in sequence alignment, we only included the conserved Trx-like domain, and ambiguously aligned regions were manually removed before the phylogenetic analysis. The final alignment included 597 nucleotide sites. All alignments are available at Fig. Share (https://figshare.com/s/18f43720186936a3effd). The processed polypeptide sequences of NCP and RCB paralogues were used for phylogenetic tree construction. The best substitution model and partition scheme were inferred using PartitionFinder v1.1.0[71]. The best maximum-likelihood tree was generated using RAxML version 7.2.8[47] with substitution model GTRGAMMAI, and multiparametric bootstrapping was conducted using RAxML with 1000 replicates. The phylogenetic tree generated was visualized in FigTree v.1.4.2, and Adobe Photoshop CC software (Adobe Systems, Inc., San Jose, CA) was used to label the name of each species.

**Transient expression in N. benthamiana.** Fluorescent-protein-tagged NCP was transiently expressed in N. benthamiana leaves. The pCHF3-NCP-CFP-FLAG plasmid was transformed by electroporation into Agrobacterium tumefaciens strain GV3101. Agrobacterium cells were grown overnight, pelleted, and resuspended in a volume of infiltration buffer equal to half the volume of the original culture. The infiltration buffer contained 10 mM MgCl2, 10 mM MES pH 5.7, and 200 μM acetosyringone (4′-hydroxy-3′,5′-dimethosyacetophenone). Cells were diluted to the O.D.$_{600}$ of 1.0. Cells were infiltrated into the abaxial side of Nicotiana benthamiana leaves. Samples were collected at 72 h after infiltration and stained with DAPI for microscopic analyses.

**Confocal imaging and quantification of photobody morphology.** For the quantification of photobody morphology, seedlings were mounted on Superfrost slides (VWR, Radnor, PA) using ddH2O and 22 × 40 mm coverslips (no. 1.5, VWR, Radnor, PA). The nuclei from the epidermal cells of the top third of the hypocotyl were imaged using a Zeiss LSM 510 inverted confocal microscope (Carl Zeiss, Thornwood, NY). GFP signal was detected using a 100× Plan-Apochromat oil immersion objective, 488 nm excitation from an argon laser, and a 505–550 nm bandpass filter. Images were collected using LSM 510 software version 4.2. Images were processed using Adobe Photoshop CC software (Adobe Systems, Inc., San Jose, CA). To determine the size and number of photobodies, the volume of photobodies was calculated using the object analyzer tool in Huygens Essentials (Scientific Volume Imaging, The Netherlands). For each nucleus, the information on the photobodies and box-and-whisker plots was sorted and calculated using Graphpad Prism 7. NCP-CFP-FLAG was detected using 458 nm excitation from an argon laserand a 470–500 nm bandpass filter.

**Nuclear and chloroplast fractionation.** For chloroplast fractionation, 2-d-old NCP-HA-His seedlings grown in Rc were frozen and homogenized in liquid nitrogen. One gram of seedlings was extracted in 2 ml of cold grinding buffer (GB, 50 mM HEPES-KOH pH 7.3, 0.33 M sorbitol, 0.1% BSA, 1 mM MnCl2, 2 mM EDTA, and 1× protease inhibitor cocktail (Millipore Sigma, St. Louis, MO)[19]. The plant extract was filtered through two layers of Miracloth (Millipore Sigma, St. Louis, MO) and centrifuged for 2 min at 2600 × g to spin down the chloroplasts. The crude chloroplasts were resuspended in 0.2 mL of GB buffer and fractionated on a Percoll (Millipore Sigma, St. Louis, MO) step gradient (80 and 40%) by centrifugation for 10 min at 2600 × g. Intact chloroplasts were obtained from the interface between the 80 and 40% Percoll.

For nuclear fractionation, seedlings were frozen in liquid nitrogen and homogenized with nuclei extraction buffer containing 20 mM PIPES-KOH pH 7.0, 10 mM MgCl2, 12% hexylene glycol, 0.25% Triton X-100, 5 mM β-mercaptoethanol, and 1× protease inhibitor cocktail. The lysate was filtered through two layers of Miracloth. The filtered lysate was loaded on top of 2 ml of 30% Percoll in 5 mM PIPES-KOH pH 7.0, 10 mM MgCl2, 3% hexylene glycol, 0.25% Triton X-100, and 5 mM β-mercaptoethanol, and centrifuged at 700 × g for 5 min at 4 °C. The fractionated nuclear pellet was dissolved in nuclei extraction buffer. Protein extracts from the chloroplast and nuclear fractions were resolved via SDS-PAGE and analyzed by immunoblot.

**Protein extraction and immunoblot analysis.** For PIF1 and PIF3 protein extraction, seedlings were ground directly in extraction buffer in a 1:3 (mg/μl) ratio, boiled for 10 min and then centrifuged at 15,000 × g for 10 min at room temperature[55]. The extraction buffer consisted of 100 mM Tris-HCl, pH 7.5, 100 mM NaCl, 5 mM EDTA, pH 8.0, 5% SDS, 20% glycerol, 20 mM dithiothreitol (DTT), 40 mM β-mercaptoethanol, 2 mM PMSF, 1× protease inhibitor cocktail, 80 μM MG132 (Millipore Sigma, St. Louis, MO), 80 μM MG115 (Millipore Sigma, St. Louis, MO), 1% phosphatase inhibitor cocktail 3 (Millipore Sigma, St. Louis, MO), and 10 mM N-ethylmaleimide. Protein extracts were separated on an SDS-PAGE mini-gel and transferred onto a polyvinylidene difluoride (PVDF) membrane. The membrane was blocked in 2% non-fat milk in 1× TBS, probed with the indicated primary antibodies, and then incubated with anti-rabbit or anti-mouse secondary antibodies conjugated with horseradish peroxidase. For the fractionation experiments, in vitro translated NCP-HA-His proteins were produced using the TNT T7-

Coupled Reticulocyte Lysate System (Promega) according to the manufacturer's protocol and detected using mouse anti-HA antibodies (11583816001, Millipore Sigma, St. Louis, MO). The purity of the chloroplast and nuclear fractions was monitored using antibodies against chloroplast ferredoxin:sulfite reductase (SiR)[72] and mouse monoclonal anti-RNA polymerase II (Pol II) antibodies (8WG16, Biolegend, San Diego, CA), respectively. Both anti-SiR and anti-Pol II antibodies were used at a 1:1000 dilution. Rabbit anti-PIF1[19] and anti-PIF3[19] polyclonal antibodies were used at a 1:500 dilution[19]. Anti-RPN6 antibodies (BML-PW8370–0100, Enzo Life Sciences, Farmingdale, NY) were used at a 1:1000 dilution. For blue native gels, mouse monoclonal anti-rpoB (PHY1700, PhytoAB Inc., Redwood City, CA) and rabbit polyclonal anti-HMR[19] antibodies were used at a 1:1000 dilution to detect the PEP complex. Secondary antibodies including anti-mouse (1706516, Bio-Rad, Hercules, CA) and anti-rabbit (1706515, Bio-Rad, Hercules, CA) were used at a 1:5000 dilution. The signals were detected with a chemiluminescence reaction using SuperSignal West Dura Extended Duration Chemiluminescent Substrate (ThermoFisher Scientific, Waltham, MA).

**Protein purification and NMR spectroscopy.** The DNA fragment encoding the Trx-like domain of NCP (residues 208–350) was PCR-amplified and cloned into a modified pET15b vector to yield an N-terminally His10-tagged protein with a TEV site between the His10-tag and the thioredoxin-like domain of NCP. The His10-tagged NCP was overexpressed in the BL21 (DE3) STAR E. coli strain (Thermo-Fisher Scientific, Waltham, MA). The cultures were grown at 37℃ until the O.D.$_{600}$ reached 0.6–0.8. The cells were induced with 0.6 mM IPTG at 20 °C for 20 h. The harvested cells were purified via Ni$^{2+}$-NTA affinity chromatography. After TEV digestion to remove the His10-tag, NCP was further purified by size-exclusion chromatography (Superdex 75, GE Healthcare Life Sciences, Marlborough, MA). Isotopically enriched proteins were overexpressed in M9 media using $^{15}$N-NH4Cl and $^{13}$C-glucose as the sole nitrogen and carbon sources (Cambridge Isotope Laboratories, Tewksbury, MA). The final NMR samples contained ~1.2 mM protein in a buffer containing 25 mM HEPES, 50 mM KCl pH 7.0, and 10 mM DTT in either 90% H2O/10% D2O or in 100% D2O.

NMR experiments were conducted using Agilent INOVA 600 or 800 MHz spectrometers at 35 °C[73]. Backbone and side-chain resonances were assigned based on standard three-dimensional triple-resonance experiments and sparsely sampled high-resolution 4D HCCH-TOCSY and HCCONH TOCSY experiments. Distance constraints were derived from 3D $^{15}$N-NOESY, sparsely sampled 4D $^{13}$C-HMQC-NOESY-$^{15}$N-HSQC and 4D $^{13}$C-HMQC-NOESY-$^{13}$C-HSQC experiments. The NMR data were processed with NMRPipe and analyzed with SPARKY[74,75]. TALOS+ analysis was used to derive the dihedral angle restraints, and CYANA was used to calculate the structures[76,77]. The final NCP ensemble of 20 structures displayed no NOE violations >0.3 Å and no dihedral angle violations >3°. The quality of the NMR ensemble was evaluated by MolProbity[78].

**Thioredoxin activity assay.** Trx activity assays were carried out in reaction buffer containing 0.1 M potassium phosphate pH 7.0, 2 mM EDTA, 0.5 mM DTT, 0.167 mM insulin as the substrate, and 1 μM His-tagged NCP or His-tagged RCB protein fragments[53]. E. coli Trx (Millipore Sigma, St. Louis, MO) and BSA (Millipore Sigma, St. Louis, MO) were used as positive and negative controls, respectively. The reaction mixtures were incubated with DTT for 5 min, and the reaction was started by adding insulin. The reduction of insulin generated turbidity, which was detected by measuring the absorbance at 650 nm every min for 1 h using a DU730 Life Science UV/Vis Spectrophotometer (Beckman Coulter, Inc., Brea, CA).

**Blue native gel electrophoresis.** Seedlings grown under the indicated conditions were ground in liquid nitrogen and resuspended in 3 volumes of BN-Lysis buffer (100 mM Tris-Cl, pH7.2; 10 mM MgCl2; 25% glycerol; 1% Triton X-100; 10 M NaF; 5 mM β-mercaptoethanol; 1× protease inhibitor cocktail)[79]. The protein extracts were divided to two tubes: one for blue-native PAGE and the other for SDS-PAGE. For blue-native PAGE, the protein extracts were mixed with BN-Sample buffer (1× NativePAGE Sample Buffer, 50 mM 6-aminocaproic acid, 1% n-dodecyl β-D-maltoside, and benzonase) and incubated for 60 min at room temperature to degrade DNA/RNA and further solubilize the PEP complex. Samples were mixed with 0.25% NativePAGE Coomassie blue G-250 Sample Additive and centrifuged at 17,500 × g for 10 min at 4 °C. Proteins from the supernatant were separated on a 4–16% NativePAGE Bis-Tris Protein Gel (ThermoFisher Scientific, Waltham, MA) according to the manufacturer's protocol (ThermoFisher Scientific, Waltham, MA) and with the following modifications. NativeMark Unstained Protein Standard (ThermoFisher Scientific, Waltham, MA) was used to determine protein size in blue-native PAGE. Briefly, electrophoresis was performed at a constant 30–40 V for 3 h at 4 °C until the blue dye migrated through one third of the gel. The Dark Blue Cathode Buffer was replaced with Light Blue Cathode Buffer, and electrophoresis continued at a constant 20–25 V overnight (16–18 h) at 4 °C. After electrophoresis was complete, the separated proteins were transferred onto a PVDF membrane using 1× NuPAGE Transfer Buffer (ThermoFisher Scientific, Waltham, MA) at a constant 70 V for 7 h at 4 °C. After transfer, the membrane was fixed with fixation buffer (25% methanol, 10% acetic acid) for 15 min and washed with water. The membrane was incubated with methanol for 3 min to destain the dye, and then immunoblotting proceeded. To determine total

amount of rpoB and HMR proteins, samples were mixed with 1× SDS Laemmli buffer containing 10% SDS and 50 mM 6-aminocaproic acid, 100 mM DTT, and 20 mM beta-mercaptoethanol, immediately boiled for 10 min, and then centrifuged at $17,500 \times g$ for 10 min at room temperature. Proteins from the supernatant were separated via SDS–PAGE and analyzed by immunoblot.

**Reporting summary**. Further information on research design is available in the Nature Research Reporting Summary linked to this article.

## Data availability

*Arabidopsis* mutants and transgenic lines, as well as plasmids generated during the current study, are available from the corresponding author upon reasonable request. The NMR assignments and the coordinate of the Trx-like domain of NCP has been deposited to Biological Magnetic Resonance Bank (30551) and RCSB Protein Data Bank (PDB ID: 6NE8), respectively. The source data underlying Figs. 1b, 1e, 1f, 2b, 2d, 2f, 2h, 2j, 3b, 3d, 3e, 4b–d, 5c, and 6g and Supplementary Figs. 1b, 1c, 2b, 2d, 3b, and 3c are provided as a Source Data file.

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

## Acknowledgements

We thank Akira Nagatani for providing the *PBG* line, Clark Lagarias for the *YHB* line, and Peter Quail for the *pifq* mutant, Sabine Heinhorst for anti-SiR antibodies. We thank Yongjian Qiu and Elise Pasoreck for helpful discussions and assistance with the paper. We thank Charles Delwiche and Caren Chang for valuable suggestions regarding the phylogenetic analysis of NCP and RCB. We thank Joanne Chory and Sabeeha Merchant for suggestions about naming of the *ncp* mutant. This work was supported by the National Institute of General Medical Sciences grant R01GM087388 and the National Science Foundation grant IOS-1051602 to M.C.

## Author contributions

E.J.Y., C.Y., J.L., P.Z. and M.C. conceived the original research plan. M.C., P.Z., T.P., D.W. and K.M.P. supervised the experiments. E.J.Y., C.Y., J.L., H.W. and J.C. performed the experiments. E.J.Y., C.Y., J.L., H.W., J.C., P.Z. and M.C. analyzed the data. J.C. and D.W. carried out the SHOREmap experiments. E.J.Y., F.L. and K.M.P. performed the phylogenetic analysis. E.J.Y., C.Y., P.Z. and M.C. wrote the article with contributions from all the authors.

## Additional information

**Competing interests:** The authors declare no competing interests.

