## [Peer Review File · Nature Communications]

Reviewers' comments:

Reviewer #1 (Remarks to the Author):

The manuscript by Yang et al describes an interesting mechanism of chloroplast transcription by RCBL. My comments are only related to the structural biology part of the manuscript.

The authors suggest that the C-terminal part of RCBL folds into a thioredoxin-like domain. Since no biological function has been ascribed to this domain, it is not clear why the authors chose to solve the structure of this domain in the first place.

The presented solution structure of RCBL is visibly similar to *E. coli* Trx. Unfortunately, it is impossible to verify this claim since the raw data are not available.

1. The authors need to present an ¹⁵N-HSQC spectrum of RCBL with assigned peaks.
2. They also need to show the assignments of the secondary structure based on both chemical shift indices and the NOE patterns of alpha helices and beta sheets.
3. The NMR data have to be deposited to the BMRB bank and the coordinates into the PDB bank.

Reviewer #2 (Remarks to the Author):

Yang et al performed a forward genetic screen for the combination of tall and albino seedling phenotypes. The authors used the PBG (PHYB-GFP) background and identified Regulator of Chloroplast Biogenesis by Light (RCBL) as a necessary component of phytochrome signaling for PhAPG activation. RCBL is shown to be dual-targeted to plastids and the nucleus. It promotes the localization of phytochrome B to photobodies where it is involved in the degradation of PIF1 and PIF3. In parallel, RCBL in plastids facilitates the assembly of the PEP into a 1000-kDa complex for PhAPG transcription.

The work by Yang et al., addresses an interesting question and the results presented provides an important step towards understanding the interaction between the nucleus and the plastids during the process of chloroplast development. The work is thorough and the experiments are professionally performed. The text is nicely written and the figures are well presented. I have however some questions that should be addressed to highlight the novelty of the dual localization of RCBL.

Major comments:

1. The major investigator of this study has previously described HEMERA, HMR (PTAC12), as a dually localized protein involved both in phytochrome signaling and plastid transcription. My major question is how does this newly identified protein, RCBL relate to the action of HMR?? As far as I can understand they seem to have quite similar functions.
2. Timing must be critical here, what is the timing of RCBL localization during the early light response? The data suggest RCBL is first imported and processed in the chloroplast to later be translocated to the nucleus. Does this mean RCBL is present in the plastids in the dark? Is the translocation from the plastids light triggered? This needs to be determined. Deletion variants without the plastid transit peptide could be used to determine whether RCBL would enter the nucleus without coming from the chloroplast.
3. Fig. 4 the immunoblots showing the level of the PEP complex (blue-native PAGE) in 4-d-old Col-0, pifq, rcbl-10, and rcbl-10/pifq seedlings grown in 10 μ mol m⁻² s⁻¹ red light. It would be informative to see if the complex would assemble in the dark in the pifq mutant to determine if the role of RCBL on PEP assembly is dependent on RCBL nuclear action.
4. The different roles of RCBL and RCB is unclear and should be developed. The reference cited

regarding RCB is not available.

Minor comments:

1. More information about the PBG line would be useful for the reader.
2. More information should be provided about the Y276H mutant of phyB (YHB).
3. The method for the in vitro translation is missing in the manuscript.
4. Ref# 57. Yoo, C. et al. "Control of chloroplast transcription by phytochrome signaling" is not to be found?

Reviewer #3 (Remarks to the Author):

In this work, Yang and co-authors identify RCBL as a new regulatory component of the expression of photosynthesis-associated plastid-encoded genes (PhAPGs). Previous studies had identified RCBL (also named MRL7-L or SVR4-like) as a nuclear-encoded protein localized in the stroma of the chloroplast, involved in the proper function of the plastid transcriptional machinery and chloroplast biogenesis. Here, Yang and co-authors confirm these observations and also provide evidence that RCBL facilitates the assembly and activation of the plastid-encoded RNA polymerase (PEP) complex. Moreover, authors show that RCBL is also present in the nucleus, and that lack of RCBL in *rcbl* mutants results in incomplete degradation of the nuclear PIF transcription factors, resulting in long-hypocotyl phenotypes. Genetic analyses show that the elevated PIF levels in the *rcbl* mutant are responsible for the long-hypocotyl phenotype, but are not directly involved in establishing the assembly and activation of the PEP complex neither the albino phenotype. This is in contrast to the closest paralogue RCB, which mediates the assembly and activation of the PEP complex exclusively through the down-regulation of PIF levels, as reported by the authors in a paper submitted elsewhere (Yoo et al.)

The question is interesting and novel, and, overall, experiments appear well-performed and data well-analyzed. Together with the related paper on RCB by Yoo et al., the manuscript describes a new regulatory connection between nuclear and chloroplast genomes, and represents an important advance towards understanding the mechanisms of light and phytochrome control of chloroplast biogenesis. Although the work describes important mechanistic insights, i.e. the observation that RCBL is required for PIF-degradation and for the assembly of the PEP complex, no detailed molecular mechanisms are provided. For example, the question of whether RCBL associates with the PEP complex in the chloroplast or with the PIF-degradation machinery in the nucleus is not addressed in the current paper. There are also some other concerns about the data that authors should address:

(1) Based on the overexpression of a RCBL-CFP-FLAG fusion protein in tobacco cells and a RCBL-HA-His fusion protein in transgenic plants, authors conclude both by confocal microscopy and by fractionation experiments that RCBL is dually localized in the nucleus and in the chloroplast. However, the alternative interpretation, that the observed nuclear localization is the result of an artifact of the overexpression is not contrasted. In the absence of antibodies against the native protein, it would be very interesting to test the localization and activity of mutant variants of the protein lacking the transit peptide or the nuclear localization signal.

(2) Also, it is intriguing that the overexpressed protein that accumulates in the nuclear fraction has a molecular weight consistent with its processed form (i.e. lacking the transit peptide). Furthermore, in vitro translation of the full-length cDNA yields a band corresponding to the full-

length protein and a lower molecular weight band, of a size similar to the protein without the transit peptide. I wonder about the origin of this second lower band, whether it might be a product of degradation, or generated by the use of an alternative ATG start site that could also operate in vivo.

(3) It has been reported that chloroplast retrograde signals contribute to seedling photomorphogenesis (Martín et al., 2016, Nature Communications). In particular, chloroplast malfunctioning induced chemically or genetically causes a reduction of the cotyledon aperture and a partially elongated hypocotyl phenotype, even in the *pifq* mutant lacking four PIFs. Interestingly, whereas the *pifq* mutation certainly suppressed most of the *rcbl-10* mutant elongated phenotype (Figure 4a), it seems that the quintuple mutant *rcbl-10/pifq* is slightly but significantly taller than the *pifq* mutant. This partial elongated phenotype of *rcbl-10/pifq* is therefore independent of the increased PIF levels observed in the *rcbl-10* mutant. Authors should discuss about this phenotype and acknowledge the contribution of the above-mentioned paper.

(4) In Figure 4a, the albino phenotypes are difficult to observe. Can authors provide a magnified image to show larger cotyledons?

(5) Authors claim "Intriguingly, the two well characterized light-labile PIFs, PIF1 and PIF3, failed to be degraded in light-grown *rcbl-1/PBG* and *rcbl-10* (Fig. 3e)."

I think this is an over interpretation of the data because authors do not analyze PIF levels in the dark and therefore, incomplete PIF degradation cannot be ruled out. Moreover, authors claim that RCBL is an early component of phytochrome signaling. However, authors only analyze long-term responses (4 days in red light). I think it would be very relevant to study the PIF levels during early dark-to-light transition (i.e. within 1h of illumination).

(6) Figure 2b and 2c: Phenotypic data are normalized to dark conditions. To have the complete picture, dark data must be shown, at least in the supplemental material.

(7) Supplemental figure 2: Authors claim that no phenotype is observed in white light or in blue light to conclude that RCBL does not participate in cryptochrome signaling. However, it is well reported that *phyB* mutants are partially tall in white light, and it is therefore somewhat surprising that *rcbl* mutants do not show a long hypocotyl in white light. The absence of phenotype might be due to the saturating light conditions used (33 $\mu\text{mol}/\text{m}^2 \text{ s}$), or to the activation of other photoreceptors that suppresses the *rcbl* mutant phenotype.

(8) Can the authors indicate in Figure 1C or in Supplemental Figure 1 the location of the primers used to detect RCBL expression, compared to the positions of *rcbl-10* and *rcbl-1* mutations? It is intriguing that the single point mutation *rcbl-1* also causes a reduction in RCBL transcript levels.

(9) Typo in lines 526 and 527 (missing °C?) "The cultures were grown at "37" until the O.D.600 reached 0.6-0.8. The cells were induced with 0.6 mM IPTG at "20" for 20 hr."

Response to Reviewers

Reviewer #1

The manuscript by Yang et al describes an interesting mechanism of chloroplast transcription by RCBL. My comments are only related to the structural biology part of the manuscript.

The authors suggest that the C-terminal part of RCBL folds into a thioredoxin-like domain. Since no biological function has been ascribed to this domain, it is not clear why the authors chose to solve the structure of this domain in the first place.

Response: We appreciate the reviewer's comment. The C-terminus of RCBL (and RCB) is predicted to form a thioredoxin-like domain based on Phyre2 homology modeling. A previous study reported that RCB displays disulfide reductase activity (Yu et al. 2014 *Mol Plant* 7:206-17). However, as the predicted thioredoxin-like domain does not have the conserved -Cys-X-X-Cys- catalytic motif, how it possesses the disulfide reductase activity has remained a mystery. We wanted to solve the structure of the thioredoxin-like domain of RCBL (the recombinant thioredoxin-like domain of RCB is unstable) to seek a possible explanation for the reductase activity. Our structural analysis of the thioredoxin-like domains of RCBL and RCB did not reveal any structural basis for the reductase activity. Consistently, we did not detect any disulfide reductase activity in RCB and RCBL. We have revised the pertinent text to clarify the motivation of the structural analysis.

The presented solution structure of RCBL is visibly similar to E. coli Trx. Unfortunately, it is impossible to verify this claim since the raw data are not available.

1. The authors need to present an 15N-HSQC spectrum of RCBL with assigned peaks.

Response: A ¹H-¹⁵N HSQC spectrum of RCBL is included in the revised manuscript as a Supplementary Fig. 4.

2. They also need to show the assignments of the secondary structure based on both chemical shift indices and the NOE patterns of alpha helices and beta sheets.

Response: In response to the reviewer request, we have included supplementary figures of the TALOS+ output, which is a neutral network secondary structure prediction based on chemical shift indices (CSI) (Supplementary Fig. 5), and a sequence plot of the observed NOE pattern (Supplementary Fig. 6). The presence of β -strands (extended conformations) is reflected by very strong NOE signals of $d_{\alpha\text{N}}(i, i+1)$ (indicated by thick bars), whereas the presence of helices is reflected by the characteristic NOEs of $d_{\alpha\beta}(i, i+3)$, $d_{\alpha\text{N}}(i, i+3)$ and $d_{\alpha\text{N}}(i, i+4)$.

3. The NMR data have to be deposited to the BMRB bank and the coordinates into the PDB bank.

Response: The NMR assignments and the coordinate of the thioredoxin-like domain of RCBL have been deposited to BMRB (30551) and RCSB (PDB ID: 6NE8), respectively.

Reviewer #2:

Yang et al performed a forward genetic screen for the combination of tall and albino seedling phenotypes. The authors used the PBG (PHYB-GFP) background and identified Regulator of Chloroplast Biogenesis by Light (RCBL) as a necessary component of phytochrome signaling for PhAPG activation. RCBL is shown to be dual-targeted to plastids and the nucleus. It promotes the localization of phytochrome B to photobodies where it is involved in the degradation of PIF1 and PIF3. In parallel, RCBL in plastids facilitates the assembly of the PEP into a 1000-kDa complex for PhAPG transcription.

The work by Yang et al., addresses an interesting question and the results presented provides an important step towards understanding the interaction between the nucleus and the plastids during the process of chloroplast development. The work is thorough and the experiments are professionally performed. The text is nicely written and the figures are well presented. I have however some questions that should be addressed to highlight the novelty of the dual localization of RCBL.

Major comments:

1. *The major investigator of this study has previously described HEMERA, HMR (PTAC12), as a dually localized protein involved both in phytochrome signaling and plastid transcription. My major question is how does this newly identified protein, RCBL relate to the action of HMR?? As far as I can understand they seem to have quite similar functions.*

Response: Both HMR and RCBL are dual-localized to plastids and the nucleus and are required for phytochrome signaling in the nucleus and the assembly of the PEP in plastids. However, the main difference is that HMR is one of the 12 PEP-associated proteins, whereas RCBL is not one of the components of the PEP but rather a regulator of PEP assembly.

2. *Timing must be critical here, what is the timing of RCBL localization during the early light response? The data suggest RCBL is first imported and processed in the chloroplast to later be translocated to the nucleus. Does this mean RCBL is present in the plastids in the dark? Is the translocation from the plastids light triggered? This needs to be determined. Deletion variants without the plastid transit peptide could be used to determine whether RCBL would enter the nucleus without coming from the chloroplast.*

Response: One of the surprises from our genetic studies is that phytochrome signaling in the nucleus and the PEP function in plastids share common components such as HMR and RCBL. It is also unexpected that the nuclear fractions of HMR and RCBL had the same molecular mass as the corresponding plastidial fractions. We agree with this reviewer that, based on these surprising observations, it becomes important to investigate the regulation of the dual-targeting of these proteins as well as to understand the significance of the dual-localization in signaling. We have begun to look into the mechanism of dual-localization of HMR. We recently reported evidence

supporting the model that HMR localizes to the plastids first and then translocates to the nucleus (Nevarez et al. 2017 *Plant Physiol* 173:1953-66). However, the mechanism of such a plastid-to-nucleus protein translocation pathway is almost completely unknown. In HMR's case, we could detect HMR in both the nucleus and plastids in the dark (Galvao et al. 2012 *Genes Dev* 26:1851-63). As you can see, addressing these questions requires careful assessments including biochemical and genetic experiments. Therefore, we feel that these studies should be included in a subsequent study. We have added the following sentence in the discussion: "Our genetic studies have so far identified three dual-targeted nuclear/plastidial molecules in PHY signaling - HMR, RCB, and RCBL (this study). One pressing upcoming task is to determine the regulation and mechanism of their dual localization as well as to understand the significance of their dual-localization in PHY signaling and nucleus-plastid communication."

3. *Fig. 4 the immunoblots showing the level of the PEP complex (blue-native PAGE) in 4-d-old Col-0, pifq, rcbl-10, and rcbl-10/pifq seedlings grown in 10 μmol m⁻² s⁻¹ red light. It would be informative to see if the complex would assemble in the dark in the pifq mutant to determine if the role of RCBL on PEP assembly is dependent on RCBL nuclear action.*

Response: This is a great question. We have now attached to this submission a full version of the RCB manuscript (Yoo et al. under review at *Nat Commun*). In the RCB study, we showed that the *pifq* mutant allows PEP assembly and *PhAPG* activation in the dark. These results, combined with the *rcbl-10/pifq* data in this study (Fig. 4), indicate that RCBL plays an essential to promote PEP assembly in plastids, which is likely separate from its nuclear role in phytochrome-mediated PIF degradation, because the *rcbl-10/pifq* mutant did rescue the long hypocotyl phenotype of *rcbl-10* (Fig. 4a).

4. *The different roles of RCBL and RCB is unclear and should be developed. The reference cited regarding RCB is not available.*

Response: RCB and RCBL are paralogous proteins playing non-redundant roles in the same nucleus-to-plastid signaling pathway (Fig. 5b, c). As shown in the attached manuscript on RCB, the main difference between RCB and RCBL is that RCB activates *PhAPG* expression primarily from the nucleus by mediating PIF degradation, whereas RCBL plays dual essential roles in PIF degradation in the nucleus as well as PEP assembly in plastids.

Minor comments:

1. *More information about the PBG line would be useful for the reader.*

Response: We added the description of the *PBG* line in the Results: "The screen was conducted in the *PBG* (PHYB-GFP) background, a transgenic line in the null *phyB-5* background complemented with functional PHYB-GFP³¹. This screening strategy allowed us to assess whether the early signaling event of photobody formation is impaired in the mutants¹⁹."

2. *More information should be provided about the Y276H mutant of phyB (YHB).*

Response: We have added the following description of the *YHB* line in the Results: “To further demonstrate RCBL’s role in PHY signaling, we crossed *rcbl-1* to a constitutively-active *phyB* allele *YHB*, which carries a Y276H mutation in PHYB’s photosensory chromophore attachment domain that locks PHYB in an active form³⁸.”

3. *The method for the in vitro translation is missing in the manuscript.*

Response: We have added the *in vitro* translation and plasmid information in the Methods.

4. *Ref# 57. Yoo, C. et al. “Control of chloroplast transcription by phytochrome signaling” is not to be found?*

Response: We have now included a full version of the manuscript on RCB, which is also under review at *Nat Commun*.

Reviewer #3:

*In this work, Yang and co-authors identify RCBL as a new regulatory component of the expression of photosynthesis-associated plastid-encoded genes (PhAPGs). Previous studies had identified RCBL (also named MRL7-L or SVR4-like) as a nuclear-encoded protein localized in the stroma of the chloroplast, involved in the proper function of the plastid transcriptional machinery and chloroplast biogenesis. Here, Yang and co-authors confirm these observations and also provide evidence that RCBL facilitates the assembly and activation of the plastid-encoded RNA polymerase (PEP) complex. Moreover, authors show that RCBL is also present in the nucleus, and that lack of RCBL in *rcbl* mutants results in incomplete degradation of the nuclear PIF transcription factors, resulting in long-hypocotyl phenotypes. Genetic analyses show that the elevated PIF levels in the *rcbl* mutant are responsible for the long-hypocotyl phenotype, but are not directly involved in establishing the assembly and activation of the PEP complex neither the albino phenotype. This is in contrast to the closest paralogue RCB, which mediates the assembly and activation of the PEP complex exclusively through the down-regulation of PIF levels, as reported by the authors in a paper submitted elsewhere (Yoo et al.)*

The question is interesting and novel, and, overall, experiments appear well-performed and data well-analyzed. Together with the related paper on RCB by Yoo et al., the manuscript describes a new regulatory connection between nuclear and chloroplast genomes, and represents an important advance towards understanding the mechanisms of light and phytochrome control of chloroplast biogenesis. Although the work describes important mechanistic insights, i.e. the observation that RCBL is required for PIF-degradation and for the assembly of the PEP complex, no detailed molecular mechanisms are provided. For example, the question of whether RCBL associates with the PEP complex in the chloroplast or with the PIF-degradation machinery in the nucleus is not addressed in the current paper. There are also some other concerns about the data that authors should address:

(1) *Based on the overexpression of a RCBL-CFP-FLAG fusion protein in tobacco cells and a RCBL-HA-His fusion protein in transgenic plants, authors conclude both by confocal microscopy and by fractionation experiments that RCBL is dually localized in the nucleus and in the chloroplast. However, the alternative interpretation, that the observed nuclear localization is the result of an artifact of the overexpression is not contrasted. In the absence of antibodies against the native protein, it would be very interesting to test the localization and activity of mutant variants of the protein lacking the transit peptide or the nuclear localization signal.*

Response: We thank the reviewer for this comment. We agree that it would be better if we could present localization data for endogenous RCBL. We have tried to generate antibodies against both RCB and RCBL. However, the antibodies for RCBL did not work. We provided in the RCB manuscript that endogenous RCB localizes to both the nucleus and plastids. We have added the following in the discussion:

“The dual localization of RCBL is supported by transiently expressed RCBL-CFP-FLAG (Fig. 3a) and subcellular fractionation results using a *RCBL-HA-His/rcbl-10* transgenic line (Fig. 3b). Although it is possible that the nuclear localization of RCBL in these experiments could be due to overexpression of RCBL, this is highly unlikely because a direct role of RCBL in nuclear PHY signaling is also supported by the overwhelming genetic evidence. RCBL is required for both PHYA and PHYB signaling (Fig. 2). RCB participates in the early light signaling events of photobody biogenesis (Fig. 3c-d). Moreover, both PIF1 and PIF3 accumulate in *rcbl-10* in the light (Fig. 3e), and the long hypocotyl phenotype of *rcbl-10* was rescued in *rcbl-10/pifq* mutant (Fig. 4a), further supporting the notion that RCBL is directly involved in PIF degradation in the nucleus.”

We also agree that it would very interesting to examine different versions of RCBL without either the transit peptide or the nuclear localization signal. Please see our response to the second question of Reviewer #2. We have recently reported a similar study on HMR (Nevarez et al. 2017 *Plant Physiol* 173:1953-66). As you can see, that these experiments are not trivial and require careful assessments including both biochemical and genetic experiments. We feel that those experiments is not within the scope of this manuscript and should be included a subsequent study.

(2) *Also, it is intriguing that the overexpressed protein that accumulates in the nuclear fraction has a molecular weight consistent with its processed form (i.e. lacking the transit peptide). Furthermore, in vitro translation of the full-length cDNA yields a band corresponding to the full-length protein and a lower molecular weight band, of a size similar to the protein without the transit peptide. I wonder about the origin of this second lower band, whether it might be a product of degradation, or generated by the use of an alternative ATG start site that could also operate in vivo.*

Response: It is very common that an *in vitro* transcription/translation experiment using the TNT kit produces a smaller band besides the expected full-length band. This could be due to either

alternative translational start site or incomplete translation. In the case of RCBL, there is a second Met in residue 56, so it is possible that some *in vitro* translation reactions started at this site. We have focused only on the full-length bands with the expected molecular masses, because we intended to use the full-length bands as size markers.

(3) *It has been reported that chloroplast retrograde signals contribute to seedling photomorphogenesis (Martín et al., 2016, Nature Communications). In particular, chloroplast malfunctioning induced chemically or genetically causes a reduction of the cotyledon aperture and a partially elongated hypocotyl phenotype, even in the pifq mutant lacking four PIFs. Interestingly, whereas the pifq mutation certainly suppressed most of the rcbl-10 mutant elongated phenotype (Figure 4a), it seems that the quintuple mutant rcbl-10/pifq is slightly but significantly taller than the pifq mutant. This partial elongated phenotype of rcbl-10/pifq is therefore independent of the increased PIF levels observed in the rcbl-10 mutant. Authors should discuss about this phenotype and acknowledge the contribution of the above-mentioned paper.*

Response: We appreciate this suggestion. We have added the reference and revised the text:

“The *rcbl-10/pifq* mutant was slightly but significantly taller than *pifq* (Fig. 4a, b), which could be due to RCBL-dependent regulation of other PIFs or a PIF-independent retrograde signaling from the defective chloroplasts, as *rcbl-10/pifq* remained albino⁴⁶.”

(4) *In Figure 4a, the albino phenotypes are difficult to observe. Can authors provide a magnified image to show larger cotyledons?*

Response: We appreciate this comment. We have added magnified images to Fig. 4a.

(5) *Authors claim “Intriguingly, the two well characterized light-labile PIFs, PIF1 and PIF3, failed to be degraded in light-grown rcbl-1/PBG and rcbl-10 (Fig. 3e).” I think this is an over interpretation of the data because authors do not analyze PIF levels in the dark and therefore, incomplete PIF degradation cannot be ruled out. Moreover, authors claim that RCBL is an early component of phytochrome signaling. However, authors only analyze long-term responses (4 days in red light). I think it would be very relevant to study the PIF levels during early dark-to-light transition (i.e. within 1h of illumination).*

Response: We have changed the sentence to: “Intriguingly, the two well-characterized light-labile PIFs, PIF1 and PIF3¹⁴, accumulated or failed to be completely degraded in light-grown *rcbl-1/PBG* and *rcbl-10* (Fig. 3e).”

Because phytochromes constantly monitor changes in ambient light, an early phytochrome signaling event does not only refer to the event occurring when seedlings first encounter light during the dark-to-light transition. Early signaling events include all signaling events that are closely associated or directly regulated by phytochromes even under continuous light or shade conditions. With this understanding, we think that photobody dynamics and PIF degradation

under continuous light are also considered as early phytochrome signaling events. Therefore, an early phytochrome signaling event can be assessed under continuous light conditions. With that said, it would be interesting to exam whether RCBL is involved in PIF degradation during the dark-to-light transition because the mechanism of PIF degradation during the dark-to-light transition might be different from that in continuous light. However, because *rcbl* is albino and seedling lethal, we do not have the homozygous *rcbl* seeds to perform the dark-to-light transition experiment.

(6) *Figure 2b and 2c: Phenotypic data are normalized to dark conditions. To have the complete picture, dark data must be shown, at least in the supplemental material.*

Response: All source data of the hypocotyl measurements in Fig. 2 are provided in the Source Data file.

(7) *Supplemental figure 2: Authors claim that no phenotype is observed in white light or in blue light to conclude that RCBL does not participate in cryptochrome signaling. However, it is well reported that phyB mutants are partially tall in white light, and it is therefore somewhat surprising that rcbl mutants do not show a long hypocotyl in white light. The absence of phenotype might be due to the saturating light conditions used (33 $\mu\text{mol}/\text{m}^2 \text{ s}$), or to the activation of other photoreceptors that suppresses the rcbl mutant phenotype.*

Response: This is a very interesting point. We have observed the same phenotype in *hmr* (Chen et al. 2010 *Cell* 141:1230-40). Cryptochrome signaling functions by itself in monochromatic blue light, however, in white light it becomes phytochrome-dependent. We think that HMR and RCBL somehow can disrupt this dependency in early phytochrome signaling. Since this is not a major point of the paper, we did not elaborate on this point.

(8) *Can the authors indicate in Figure 1C or in Supplemental Figure 1 the location of the primers used to detect RCBL expression, compared to the positions of rcbl-10 and rcbl-1 mutations? It is intriguing that the single point mutation rcbl-1 also causes a reduction in RCBL transcript levels.*

Response: We have added the locations of the primers in Supplementary Fig. 1. The point mutation in *rcbl-1* generated a premature stop codon. It is a common observation that premature stop codon can also cause a dramatic decrease in mRNA level because the portion of mRNA not covered by ribosomes, or the “naked” mRNA, is considered unstable or more sensitive to nucleases.

(9) *Typo in lines 526 and 527 (missing °C?) "The cultures were grown at "37"↵ until the O.D.600 reached 0.6-0.8. The cells were induced with 0.6 mM IPTG at "20"↵ for 20 hr."*

Response: Corrected.

REVIEWERS' COMMENTS:

Reviewer #1 (Remarks to the Author):

I have no further comments. The authors satisfactorily resolved all my concerns.

Reviewer #2 (Remarks to the Author):

In the revised version of Yang et al., the authors have addressed some of the criticisms I raised on the last version of the manuscript. However, my major concerns with the work were left unanswered.

Major comments:

1. The major investigator of this study has previously described HEMERA, HMR (PTAC12), as a dually localized protein involved both in phytochrome signaling and plastid transcription. My major question is how does this newly identified protein, RCBL relate to the action of HMR?? As far as I can understand they seem to have quite similar functions. Response: Both HMR and RCBL are dual-localized to plastids and the nucleus and are required for phytochrome signaling in the nucleus and the assembly of the PEP in plastids. However, the main difference is that HMR is one of the 12 PEP-associated proteins, whereas RCBL is not one of the components of the PEP but rather a regulator of PEP assembly.

- To investigate the relationship between HMR, RBC and RBCL mutant combinations are required. Protein interaction studies would also shed light on the connection between these 3 proteins with very similar functions.

2. Timing must be critical here, what is the timing of RCBL localization during the early light response? The data suggest RCBL is first imported and processed in the chloroplast to later be translocated to the nucleus. Does this mean RCBL is present in the plastids in the dark? Is the translocation from the plastids light triggered? This needs to be determined. Deletion variants without the plastid transit peptide could be used to determine whether RCBL would enter the nucleus without coming from the chloroplast. Response: One of the surprises from our genetic studies is that phytochrome signaling in the nucleus and the PEP function in plastids share common components such as HMR and RCBL. It is also unexpected that the nuclear fractions of HMR and RCBL had the same molecular mass as the corresponding plastidial fractions. We agree with this reviewer that, based on these surprising observations, it becomes important to investigate the regulation of the dual-targeting of these proteins as well as to understand the significance of the dual-localization in signaling. We have begun to look into the mechanism of dual-localization of HMR. We recently reported evidence supporting the model that HMR localizes to the plastids first and then translocates to the nucleus (Nevarez et al. 2017 Plant Physiol 173:1953-66). However, the mechanism of such a plastid-to-nucleus protein translocation pathway is almost completely unknown. In HMR's case, we could detect HMR in both the nucleus and plastids in the dark (Galvao et al. 2012 Genes Dev 26:1851-63). As you can see, addressing these questions requires careful assessments including biochemical and genetic experiments. Therefore, we feel that these studies should be included in a subsequent study. We have added the following sentence in the discussion: "Our genetic studies have so far identified three dual-targeted nuclear/plastidial molecules in PHY signaling - HMR, RCB, and RCBL (this study). One pressing upcoming task is to determine the regulation and mechanism of their dual localization as well as to understand the significance of their dual-localization in PHY signaling and nucleus-plastid communication."

- It is critical to address this point. The dual localization has been described before and here are now two submissions describing the dual localization of two proteins similar to HMR. To provide

further understanding of the action of these proteins more information about the timing of the specific localization should be provided. The tools are already available in the laboratory as shown in Figure 3.

3. Fig. 4 the immunoblots showing the level of the PEP complex (blue-native PAGE) in 4-d-old Col-0, pifq, rcbl-10, and rcbl-10/pifq seedlings grown in 10 $\mu\text{mol m}^{-2} \text{s}^{-1}$ red light. It would be informative to see if the complex would assemble in the dark in the pifq mutant to determine if the role of RCBL on PEP assembly is dependent on RCBL nuclear action. Response: This is a great question. We have now attached to this submission a full version of the RCB manuscript (Yoo et al. under review at Nat Commun). In the RCB study, we showed that the pifq mutant allows PEP assembly and PhAPG activation in the dark. These results, combined with the rcbl1pifq data in this study (Fig. 4), indicate that RCBL plays an essential role to promote PEP assembly in plastids, which is likely separate from its nuclear role in phytochrome-mediated PIF degradation, because the rcblpifq mutant did rescue the long hypocotyl phenotype of rcbl (Fig. 4a).

- The relationship between these two manuscripts needs to be clear. Are they expected to be published back-to-back? Personally, I think the two manuscripts should be combined into one comprehensive manuscript.

4. The different roles of RCBL and RCB is unclear and should be developed. The reference cited regarding RCB is not available. Response: RCB and RCBL are paralogous proteins playing non-redundant roles in the same nucleus-to-plastid signaling pathway (Fig. 5b, c). As shown in the attached manuscript on RCB, the main difference between RCB and RCBL is that RCB activates PhAPG expression primarily from the nucleus by mediating PIF degradation, whereas RCBL plays dual essential roles in PIF degradation in the nucleus as well as PEP assembly in plastids.

- See comments to point 1 about mutant combinations.

I am satisfied with the corrections to my minor points.

Reviewer #3 (Remarks to the Author):

Minor additional comments:

Line 171: "These results indicate that RCBL is dual-targeted to plastids and the nucleus and imply that RCBL might localize to the plastids first and then translocate to the nucleus similar to HMR." Authors state that the in deep mechanistic characterization of the dual nuclear-chloroplast targeting of RCBL is out of the scope of the present manuscript. Therefore, I believe this sentence can be misleading, since the word "imply" favors the proposed scenario without providing necessary evidence.

Line 202: "The expression of PEP-dependent PhAPGs was still impaired in rcbl-10/pifq, while the expression of NEP-dependent plastidial genes was elevated (Fig. 4b)."
The citation must refer to Fig 4c, not 4b.

Line 206: "We found that the PEP failed to form a 1000-kDa complex in rcbl-10 (Fig. 4c), indicating that RCBL is required for PEP assembly. The defect in PEP assembly was not rescued in rcbl-10/pifq (Fig. 4c)."
The citation must refer to Fig. 4d, not 4c.

Figure 4 legend: there are 2 sections c, but no section d.

Reviewer #1 I have no further comments. The authors satisfactorily resolved all my concerns.

Reviewer #2: In the revised version of Yang et al., the authors have addressed some of the criticisms I raised on the last version of the manuscript. However, my major concerns with the work were left unanswered. Major comments: 1. The major investigator of this study has previously described HEMERA, HMR (PTAC12), as a dually localized protein involved both in phytochrome signaling and plastid transcription. My major question is how does this newly identified protein, RCBL relate to the action of HMR?? As far as I can understand they seem to have quite similar functions. Response: Both HMR and RCBL are dual-localized to plastids and the nucleus and are required for phytochrome signaling in the nucleus and the assembly of the PEP in plastids. However, the main difference is that HMR is one of the 12 PEP-associated proteins, whereas RCBL is not one of the components of the PEP but rather a regulator of PEP assembly. - To investigate the relationship between HMR, RCB and RCBL mutant combinations are required. Protein interaction studies would also shed light on the connection between these 3 proteins with very similar functions. Response: Although we still do not know whether NCP (RCBL was the previous name) interacts directly with HMR and RCB, we have shown genetic evidence that NCP and RCB work in the same genetic pathway of phytochrome signaling (Fig. 5b). We agree that it would be interesting to investigate whether RCB, NCP, and HMR interact with each other. But we also feel that the detailed molecular functions of NCP can be investigated in a follow-up study, as the current study has already spanned from the identification of NCP by a forward-genetic screen, to defining its functions in the anterograde signaling, to phylogenetic analysis, and to structural studies of its thioredoxin-like domain. 2. Timing must be critical here, what is the timing of RCBL localization during the early light response? The data suggest RCBL is first imported and processed in the chloroplast to later be translocated to the nucleus. Does this mean RCBL is present in the plastids in the dark? Is the translocation from the plastids light triggered? This needs to be determined. Deletion variants without the plastid transit peptide could be used to determine whether RCBL would enter the nucleus without coming from the chloroplast. Response: One of the surprises from our genetic studies is that phytochrome signaling in the nucleus and the PEP function in plastids share common components such as HMR and RCBL. It is also unexpected that the nuclear fractions of HMR and RCBL had the same molecular mass as the corresponding plastidial fractions. We agree with this reviewer that, based on these surprising observations, it becomes important to investigate the regulation of the dual-targeting of these proteins as well as to understand the significance of the dual-localization in signaling. We have begun to look into the mechanism of dual-localization of HMR. We recently reported evidence supporting the model that HMR localizes to the plastids first and then translocates to the nucleus (Nevarez et al. 2017 *Plant Physiol* 173:1953-66). However, the mechanism of such a plastid-tonucleus protein translocation pathway is almost completely unknown. In HMR's case, we could detect HMR in both the nucleus and plastids in the dark (Galvao et al. 2012 *Genes Dev* 26:1851-63). As you can see, addressing these questions requires careful assessments including biochemical and genetic experiments. Therefore, we feel that these studies should be included in a subsequent study. We have added the following sentence in the discussion: "Our genetic studies have so far identified three dual-targeted nuclear/plastidial molecules in PHY signaling - HMR, RCB, and RCBL (this study). One pressing upcoming task is to determine the regulation and mechanism of their dual localization as well as to understand the significance of their dual-localization in PHY signaling and nucleus-plastid communication." - It is critical to address this point. The dual localization has been described before and here are now two submissions describing the dual localization of two proteins similar to HMR. To provide further understanding of the action of these proteins more information about the

timing of the specific localization should be provided. The tools are already available in the laboratory as shown in Figure 3. Response: We agree with this reviewer that it is important to look into the regulation of NCP's dual localization. However, we also think that these questions should be carefully examined in future investigations for the same reasons explained above.

3. Fig. 4 the immunoblots showing the level of the PEP complex (blue-native PAGE) in 4-d-old Col-0, *pifq*, *rcbl-10*, and *rcbl-10/pifq* seedlings grown in 10 $\mu\text{mol m}^{-2} \text{s}^{-1}$ red light. It would be informative to see if the complex would assemble in the dark in the *pifq* mutant to determine if the role of RCBL on PEP assembly is dependent on RCBL nuclear action. Response: This is a great question. We have now attached to this submission a full version of the RCB manuscript (Yoo et al. under review at Nat Commun). In the RCB study, we showed that the *pifq* mutant allows PEP assembly and PhAPG activation in the dark. These results, combined with the *rcbl1pifq* data in this study (Fig. 4), indicate that RCBL plays an essential role to promote PEP assembly in plastids, which is likely separate from its nuclear role in phytochrome-mediated PIF degradation, because the *rcblpifq* mutant did rescue the long hypocotyl phenotype of *rcbl* (Fig. 4a). - The relationship between these two manuscripts needs to be clear. Are they expected to be published back-to-back? Personally, I think the two manuscripts should be combined into one comprehensive manuscript. Response: These two manuscripts represent back-to-back stories of the discovery of the nucleusto-plastid signaling and two novel phytochrome signaling components in the anterograde signaling. Although RCB and RCBL are paralogs, they play distinct roles in the anterograde signaling. We think it is appropriate to present them as separate stories.

4. The different roles of RCBL and RCB is unclear and should be developed. The reference cited regarding RCB is not available. Response: RCB and RCBL are paralogous proteins playing non-redundant roles in the same nucleus-to-plastid signaling pathway (Fig. 5b, c). As shown in the attached manuscript on RCB, the main difference between RCB and RCBL is that RCB activates PhAPG expression primarily from the nucleus by mediating PIF degradation, whereas RCBL plays dual essential roles in PIF degradation in the nucleus as well as PEP assembly in plastids. - See comments to point 1 about mutant combinations. Response: Please see our response to point 1. I am satisfied with the corrections to my minor points.

Reviewer #3: Minor additional comments: Line 171: "These results indicate that RCBL is dual-targeted to plastids and the nucleus and imply that RCBL might localize to the plastids first and then translocate to the nucleus similar to HMR." Authors state that the in deep mechanistic characterization of the dual nuclear-chloroplast targeting of RCBL is out of the scope of the present manuscript. Therefore, I believe this sentence can be misleading, since the word "imply" favors the proposed scenario without providing necessary evidence. Response: Because plastidial NCP is a mature form without its transit peptide and transit peptide is processed in plastids, the fact that nuclear and plastidial NCP proteins had the same molecular size (both are smaller than the full-length) does imply that NCP is targeted to plastids first before translocating to the nucleus, in a similar way as HMR. Line 202: "The expression of PEP-dependent PhAPGs was still impaired in *rcbl-10/pifq*, while the expression of NEP-dependent plastidial genes was elevated (Fig. 4b)." The citation must refer to Fig 4c, not 4b. Response: We thank this reviewer for this comment. We have made the change. Line 206: "We found that the PEP failed to form a 1000-kDa complex in *rcbl-10* (Fig. 4c), indicating that RCBL is required for PEP assembly. The defect in PEP assembly was not rescued in *rcbl-10/pifq* (Fig. 4c)." The citation must refer to Fig. 4d, not 4c. Response: We have made the change. Figure 4 legend: there are 2 sections c, but no section d. Response: We have made the change.